# p53 regulates diverse tissue-specific outcomes to endogenous DNA damage in mice

Ross J. Hill [1], Nazareno Bona[1], Job Smink[2], Hannah K. Webb [1], Alastair Crisp[1], Juan I. Garaycoechea [2] ✉ & Gerry P. Crossan [1] ✉

DNA repair deficiency can lead to segmental phenotypes in humans and mice, in which certain tissues lose homeostasis while others remain seemingly unaffected. This may be due to different tissues facing varying levels of damage or having different reliance on specific DNA repair pathways. However, we find that the cellular response to DNA damage determines different tissue-specific outcomes. Here, we use a mouse model of the human XPF-ERCC1 progeroid syndrome (XFE) caused by loss of DNA repair. We find that p53, a central regulator of the cellular response to DNA damage, regulates tissue dysfunction in *Ercc1-/-* mice in different ways. We show that ablation of p53 rescues the loss of hematopoietic stem cells, and has no effect on kidney, germ cell or brain dysfunction, but exacerbates liver pathology and polyploidisation. Mechanistically, we find that p53 ablation led to the loss of cell-cycle regulation in the liver, with reduced p21 expression. Eventually, *p16/Cdkn2a* expression is induced, serving as a fail-safe brake to proliferation in the absence of the p53-p21 axis. Taken together, our data show that distinct and tissue-specific functions of p53, in response to DNA damage, play a crucial role in regulating tissue-specific phenotypes.

The importance of detecting and repairing DNA damage is illustrated by the severe and life-limiting phenotypes observed when humans lack DNA repair[1,2]. Human patients with severe mutations in the XPF-ERCC1 endonuclease, present with XFE progeroid syndrome[3,4]. Seminal work has exploited XFE to uncover fundamental insights into the aging process[4–14]. This is a complex multi-organ system failure syndrome characterised by failure to thrive, cachexia, liver failure, renal insufficiency, neurodegeneration and anaemia, reminiscent of premature aging[4,9,15]. However, these phenotypes are segmental with only a subset of tissues affected, which would not be predicted from cell-based studies as, regardless of cellular origin (and species), *XPF-/-* or *ERCC1-/-* cells are invariably hypersensitive to agents that cause bulky helix-distorting adducts, intra- or inter-strand crosslinks[16]. This segmental nature of the in vivo phenotype could in part be explained by; (i) different tissues having distinct exposures to DNA damage, (ii) different

tissues utilising alternative DNA repair pathways to resolve the same damage (e.g., post-mitotic cells cannot perform replication-coupled repair), or (iii) different tissues employing alternative quality control mechanisms that result in distinct cellular outcomes following damage.

A universal consequence of DNA damage is the engagement of a cellular 'DNA damage response'[17]. The tumour suppressor p53 is of such central importance to this process that it has acquired the moniker of 'guardian of the genome'[18–20]. In response to DNA damage, p53 becomes phosphorylated and accumulates in the cell, leading to transcriptional changes that act to promote DNA repair, alter the cell cycle or promote cell death. The mechanisms that determine which of these outcomes predominates is poorly understood. However, these processes act to limit the risk of damaged or mutated genetic information being passed onto daughter cells, preserving genome stability and limiting the risk of neoplastic transformation.

[1]MRC Laboratory of Molecular Biology, Cambridge Biomedical Campus, Francis Crick Avenue, Cambridge, UK. [2]Hubrecht Institute, Royal Netherlands Academy of Arts and Sciences (KNAW), Utrecht, the Netherlands. ✉e-mail: juan.g@hubrecht.eu; gcrossan@mrc-lmb.cam.ac.uk

Therefore, p53-dependent quality control mechanisms have crucial roles in determining how tissues respond to DNA damage[21]. This essential role is highlighted by the radioprotection of organisms lacking p53[22,23]. However, p53 is known to have distinct tissue-specific transcriptional targets[24,25]. It is, therefore, plausible that differences in the balance between DNA repair, cell-cycle arrest and apoptosis will result in very different outcomes. Perhaps, this is most clearly illustrated by mouse models with persistent p53 activation, either through mutation of p53 or MDM2—a negative regulator of p53[26]. In general, tissues with high turnover are more susceptible to the loss of homoeostasis when p53 is active[27]. However, this is not exclusive to rapidly proliferating tissues e.g. liver, kidney, retina and brain also induce both apoptotic and senescent responses, albeit to a lesser extent. The complexity deepens when we take into account the gastrointestinal (GI) tract in this context. Despite having comparable proliferation to the bone marrow or thymus, p53 activity does not lead to cell loss; in fact, stabilisation of p53 or 'super-p53' both lead to radioresistance in the GI tract[28,29]. Moreover, p53 has also been shown to regulate liver ploidy and limit polyploidisation of the liver following partial hepatectomy[30]. It was also found that p21 (a transcriptional target of p53) limits liver polyploidisation[31]. This is perhaps expected given that whole genome duplication occurs only in the absence of p53 following telomere attrition[32]. However, p53 loss does not appear to be a prerequisite for whole genome duplication, as genome sequencing has shown that approximately half of whole genome duplication events happen with intact p53[33]. Conversely, a recent study found that p53 is required for genome duplication and polyploidisation by promoting mitotic bypass[34]. This shows that the relationship between p53, DNA damage and tissue responses, is more complex than initially thought.

*Ercc1*-deficient mice are a well-established model of DNA repair deficiency, XFE syndrome and premature aging[4,15]. These mice die extremely prematurely; however, the phenotypes are segmental in nature. Firstly, only a subset of tissues are affected (e.g. liver, kidney, bone marrow, brain and germ cells), whilst others remain unaltered (e.g. GI tract)[4–9,11,12]. Secondly, the affected tissues have different pathological features, e.g. haematopoietic stem cell (HSC) attrition in the bone marrow and hepatocyte polyploidy in the liver, with *Ercc1*-deficient mice rapidly succumb to liver failure. However, if the liver defect is rescued by liver-specific expression of ERCC1, then mice die from renal dysfunction[35]. Previous studies have shown that p53 is stabilised and activated in *Ercc1*-deficient mice and implicated it in regulating liver dysfunction[36,37], with p53 being involved in the regulation of both hepatocyte proliferation and binucleation in the absence of *Ercc1*[37].

Here, we comprehensively investigate the role of p53 in regulating the various tissue-specific pathological features of XFE. We find that in the absence of *Ercc1*, there is p53 stabilisation and activation of p53-transcriptional targets in several tissues. However, loss of p53 leads to different outcomes in different tissues. We find that whilst p53 loss rescues bone marrow dysfunction, it exacerbates liver polyploidisation. Mechanistically, we show that loss of p53 disables the p53-p21 axis in hepatocytes allowing increased proliferation; however, the liver becomes senescent as *p16/Cdkn2a* is activated as a failsafe brake to proliferation.

## Results

### p53 is activated in the absence of *Ercc1*

To study the relationship between endogenous DNA damage, the p53 checkpoint and tissue-specific outcomes, we employed a previously published *Ercc1*[−/−] mouse model[11,38]. We first confirmed that our model recapitulates the key pathological features of XPF-ERCC1 deficiency in mouse and human. *Ercc1*[−/−] mice were generated on a C57BL/6 x 129S4S6/Sv F1 hybrid background using the *Ercc1*[tm1a(KOMP)Wtsi] allele. From heterozygous crosses, *Ercc1*[−/−] pups were born at the expected Mendelian ratio at postnatal day 14 (P14) (24.2%, $p = 0.4470$). However,

they were severely runted compared to littermate controls, with a significant 1.8-fold reduction in body mass 21 days after birth (Supplementary Fig. 1a, b). Consistent with previous reports, *Ercc1*[−/−] mice had a significantly reduced lifespan and segmental loss of tissue homoeostasis (Supplementary Fig. 1c–f)[4,15]. Firstly, these mice had elevated blood serum levels of aspartate aminotransferase (AST) and alanine aminotransferase (ALT) activity, indicative of liver dysfunction (Supplementary Fig. 1d). Secondly, *Ercc1*[−/−] mice develop renal insufficiency with elevated blood serum levels of urea and proteinuria (Supplementary Fig. 1e, f). Finally, *Ercc1*[−/−] mice showed a significant 2.1-fold reduction in the frequency of haematopoietic stem and progenitor cells (HSPCs, lineage[-] c-Kit[+] Sca-1[+]) (Supplementary Fig. 1g)[6]. Collectively, these data demonstrate that our model recapitulates the major phenotypes of both human XFE progeroid syndrome and previously generated *Ercc1*[−/−] models.

To explore tissue differences in response to endogenous DNA damage, we initially focussed our attention on the liver, kidney, and bone marrow of *Ercc1*[−/−] mice as these are thought to contribute to reduced lifespan[4,15,35]. We stained tissues of 6-to-8-week-old *Ercc1*[−/−] and control mice for the phosphorylated histone variant H2A.X (γ-H2A.X), a marker of DNA breaks[39]. We observed a significant increase in the frequency of cells with >5 nuclear foci of γ-H2A.X in each tissue (Fig. 1a) and an increase in the average number of foci per cell (Supplementary Fig. 2a). This shows that there is a sufficient burden of endogenous genotoxin, resulting in the accumulation of unrepaired DNA damage in the absence of ERCC1 in all three tissues. Consequently, persistent DNA damage can lead to the activation of the DNA damage response. Whilst p53 is a downstream effector, it is among, if not the most, important regulator of the cellular response to DNA damage[20]. We observed a significant increase in the frequency of pSer15-p53[+] cells - a post-translational modification induced upon DNA damage—in the liver, kidney and bone marrow of *Ercc1*[−/−] mice when compared to controls (Fig. 1b). Phosphorylation and stabilisation of p53 result in the activation of a range of p53-transcriptional targets, which ultimately determine the cellular outcome[40]. One key component of this response is the activation of programmed cell death. We found that whilst all three tissues tested showed persistent DNA damage and accumulation of phosphorylated p53, only the kidney and bone marrow had an increased frequency of cells undergoing apoptosis as determined by cleaved caspase-3 staining (CC3[+]) and TUNEL assay (Fig. 1c and Supplementary Fig. 2b).

We then assessed the expression of *Cdkn1a/p21, Btg2, Mdm2, Gdf15* and *Bax* which are well-described transcriptional targets of p53 involved in different cellular processes[41]. We performed RT-qPCR on liver, kidney and HSC samples and, in agreement with the CC3 immunostaining, we observed an induction of the pro-apoptotic factor *Bax* in both kidney and blood stem cells but not in the liver (Supplementary Fig. 2c). In contrast, *Ercc1*[−/−] liver and kidney show a significant induction in the expression of the senescence marker *p21/Cdkn1a* but this is not observed in HSCs (Supplementary Fig. 2c). Together, these data show that in the absence of ERCC1, endogenous DNA damage accumulates and activates p53 in the liver, kidney, and bone marrow, but results in distinct tissue-specific transcriptional and apoptotic responses.

### Differential dependence of tissues on p53 for apoptosis activation

We then set out to genetically determine the contribution of p53 to tissue-specific responses such as apoptosis, cell-cycle arrest and polyploidisation in *Ercc1*[−/−] mice. We generated *Ercc1*[−/−]*Trp53*[−/−] mice on a C57BL/6 x 129S4S6/Sv F1 background. *Ercc1*[−/−]*Trp53*[−/−] mice have previously been generated; however, analysis was restricted to the liver and the consequences on tissue pathology and homoeostasis were not studied[37,42]. We found that *Ercc1*[−/−]*Trp53*[−/−] mice were present at a significantly reduced frequency at postnatal day P21 (3.0%, 12/395,

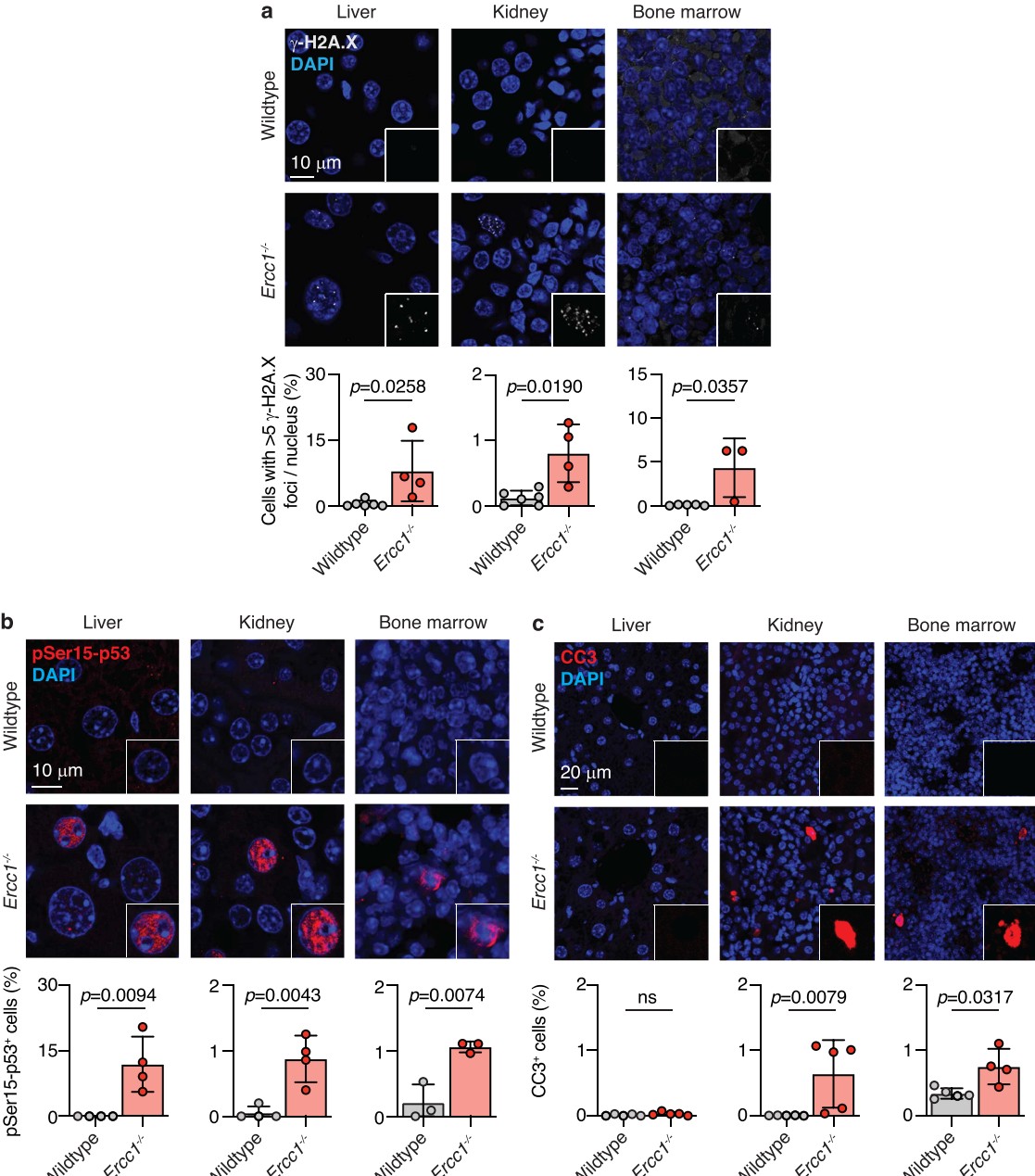

**Fig. 1 | p53 is activated in the absence of ERCC1. a** Representative images of γ-H2A.X foci in liver, kidney and bone marrow sections from wildtype and *Ercc1⁻/⁻* mice and quantification of the frequency of cells with >5 nuclear γ-H2A.X foci (*p* values calculated by two-tailed Mann–Whitney *U*-test, data were mean ± s.d.; *n* = liver (6 wildtype and 4 *Ercc1⁻/⁻*), kidney (6 wildtype and 4 *Ercc1⁻/⁻*), bone marrow (5 wildtype and 3 *Ercc1⁻/⁻*) independent mice). **b** Representative immunofluorescence images of phosphorylated TP53 (pSer15-TP53) in liver, kidney and bone marrow of wildtype and *Ercc1⁻/⁻* mice and quantification of the frequency of pSer15-TP53⁺ cells

(*p* values calculated by two-tailed Mann–Whitney *U*-test, data were mean ± s.d.; *n* = liver (4 wildtype and 4 *Ercc1⁻/⁻*), kidney (4 wildtype and 4 *Ercc1⁻/⁻*), bone marrow (3 wildtype and 3 *Ercc1⁻/⁻*) mice). **c** Representative immunofluorescence images of cleaved caspase-3 (CC3) in liver, kidney and bone marrow of wildtype and *Ercc1⁻/⁻* mice and quantification of the frequency of CC3⁺ cells (*p* values calculated by two-tailed Mann–Whitney *U*-test, data were mean ± s.d.; *n* = 5 Wildtype and 5 *Ercc1⁻/⁻*), kidney (5 wildtype and 5 *Ercc1⁻/⁻*), bone marrow (5 wildtype and 4 *Ercc1⁻/⁻*) mice). Source data are provided as a Source data file.

*p* = 0.0002; Fig. 2a). This is unexpected as *Ercc1⁻/⁻* single mutants were born at the expected ratio (Supplementary Fig. 1a). Furthermore, the disruption of p53 frequently rescues the lethality of mice deficient in a range of different DNA repair pathways rather than potentiating their phenotypes[43–50].

An obvious feature of *Ercc1*-deficient mice is a profound reduction in mass when compared to littermates, however, the aetiology of this phenotype is unknown (Fig. 2b, c, Supplementary Fig. 1b). We hypothesised that this may in part be due to high blood serum levels of the anorexic hormone growth/differentiation factor 15 (GDF15)

that we detected in high levels in the absence of ERCC1 when compared to wildtype controls (Fig. 2d)[51]. It has previously been shown that the expression of GDF15 is regulated by p53, and consistent with this, we found that the GDF15 response in *Ercc1⁻/⁻* mice was suppressed by p53 loss (Fig. 2d)[52]. Despite the suppression of GDF15 secretion, *Ercc1⁻/⁻ Trp53⁻/⁻* mice had a significant reduction in body mass that was comparable to that observed in *Ercc1⁻/⁻* controls (Fig. 2c). Hence, it is unlikely that GDF15 plays an important role in preventing weight gain in patients with XFE progeroid syndrome. The most striking feature of *Ercc1* deficiency is the dramatic reduction in

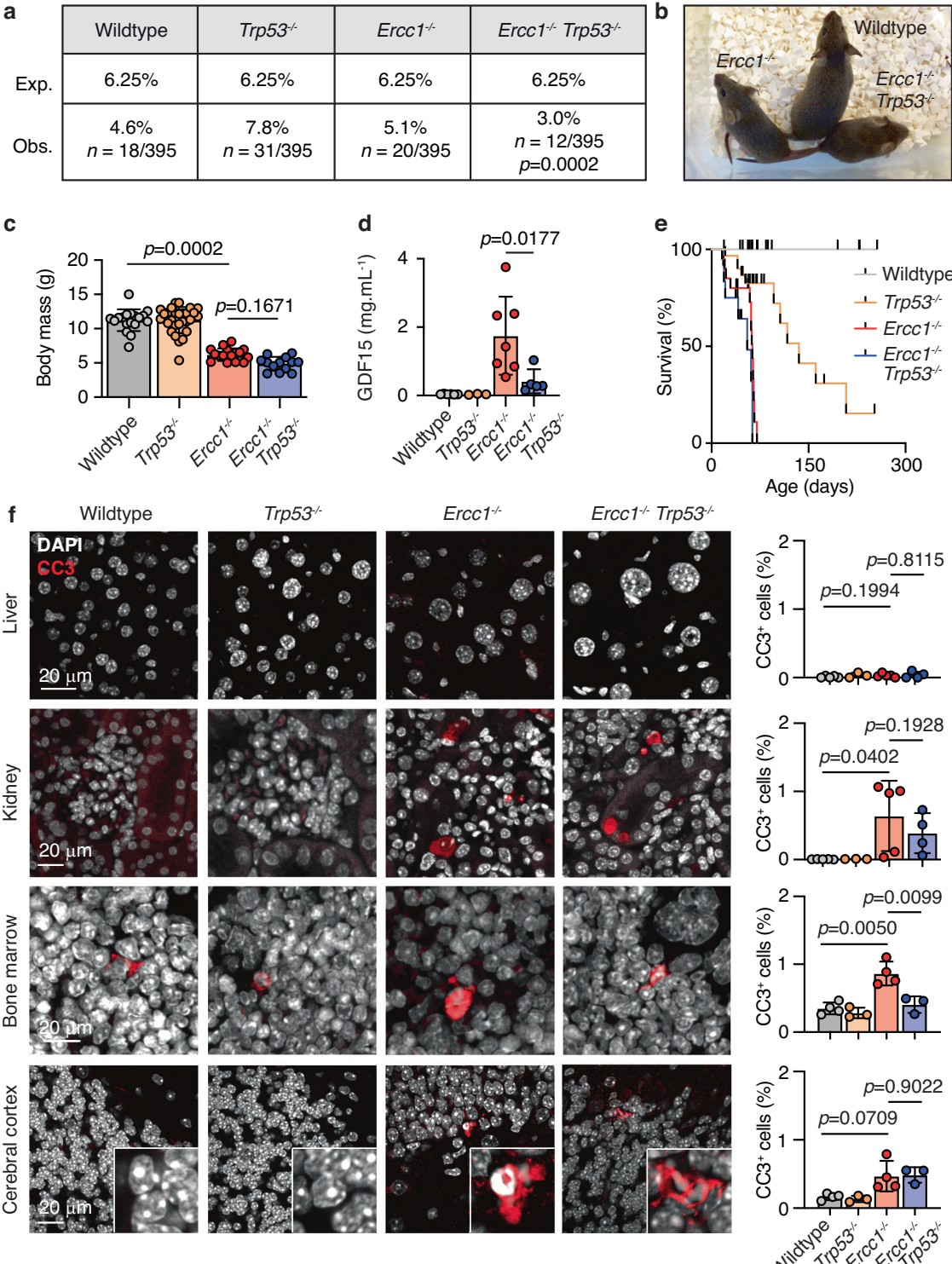

**Fig. 2 | Differential dependence of tissues on p53 for apoptosis activation.**
**a** *Ercc1⁻/⁻* mice are observed at expected Mendelian ratios at postnatal day 21 (*p* value calculated by one-tailed Chi-square test). **b** Representative photograph of 6-week-old *Ercc1⁻/⁻Trp53⁻/⁻* and littermate control mice. **c** Weight of *Ercc1⁻/⁻Trp53⁻/⁻* and control mice at postnatal day 21 (*p* values calculated by two-tailed Mann−Whitney *U*-test, data were mean ± s.d.; *n* = 16, 22, 14, 10 independent mice, left to right). **d** Terminal serum GDF15 levels from 6-to-8-week-old *Ercc1⁻/⁻Trp53⁻/⁻* and control mice (*p* value calculated by two-tailed Mann−Whitney *U*-test, data were mean ± s.d.; *n* = 6, 3, 7 and 5 independent mice, left to right). **e** Kaplan−Meier survival curve showing the survival of cohorts of *Ercc1⁻/⁻* and wildtype littermate controls (*p* value calculated by one-tailed Mantel-Cox test, *n* = 29 wildtype and 23 *Ercc1⁻/⁻* mice). **f** Left−Representative immunofluorescence of cleaved caspase-3 (CC3) staining in the liver, kidney, bone marrow and cerebral cortex of *Ercc1⁻/⁻ Trp53⁻/⁻* and control mice. Right−Quantification of the frequency of CC3⁺ cells in the liver, kidney, bone marrow and cerebral cortex of *Ercc1⁻/⁻ Trp53⁻/⁻* and control mice (*p* value calculated by two-tailed Mann−Whitney *U*-test, data were mean ± s.d.; each dot represent a mouse, Liver: *n* = 5, 3, 5, 4; Kidney: *n* = 4, 3, 5, 4; Bone marrow: *n* = 4, 3, 4, 3; Cerebral cortex: *n* = 4, 3, 4, 3 left to right). Source data are provided as a Source data file.

lifespan, however, we found that there was no significant difference in the median lifespan of *Ercc1*[-/-] *Trp53*[-/-] mice when compared to *Ercc1*[-/-] controls (55 and 63 days, respectively, *p* = 0.1224) (Fig. 2e). These data clearly show that loss of p53 does not rescue the reduced weight or longevity of *Ercc1*-deficient mice.

As we had shown that *Ercc1*-deficient tissues had increased frequency of apoptotic cells and that p53 is a regulator of apoptosis, we asked if, in the absence of p53, apoptosis was attenuated. We assessed the frequency of apoptotic cells in the liver, kidney, bone marrow, and cerebral cortex of *Ercc1*[-/-] *Trp53*[-/-] mice (Fig. 2f). In agreement with our earlier data, the liver does not activate apoptosis despite exhibiting DNA damage and increased pSer-p53 (Figs. 1a, b, 2f). We found that whilst the kidney, bone marrow and cerebral cortex of *Ercc1*[-/-] mice each had elevated frequency of apoptotic cells, the loss of p53 only led to a significant reduction in the bone marrow (*p* = 0.0099; Fig. 2f). These results suggest that p53 is the major driver of apoptosis in the bone marrow, but that response is activated by alternative mechanisms in the kidney and cerebral cortex. These data clearly show that *p53* has a complex, unpredicted and tissue-specific interaction with *Ercc1*.

## Kidney, neurological and germ cell pathologies persist in the absence of *p53*

The kidney is a vital organ which becomes dysfunctional in the absence of ERCC1-mediated DNA repair[35]. Whilst we found that loss of p53 did not rescue the apoptotic response in this organ, we examined the kidney in more detail. Firstly, we asked if loss of p53 exacerbated the burden of DNA damage in *Ercc1*[-/-] kidney, as it could explain why the loss of p53 did not affect the frequency of apoptotic cells. We found that the loss of p53 did not alter the frequency of cells with markers of DNA damage when compared to *Ercc1*[-/-] littermates (Supplementary Fig. 3a). Next we performed RT-qPCR on kidney samples with a panel of p53-transcriptional targets involved in different responses. We measured the pro-apoptotic factors *Bax* and *Fas*, and the cell-cycle/proliferation regulator *p21/Cdkn1a* and *Btg2*[41]. We found that they were all upregulated in both *Ercc1*[-/-] and *Ercc1*[-/-] *Trp53*[-/-] kidneys (Fig. 3a). The observation that loss of p53 did not reduce the expression of the pro-apoptotic factors *Bax* and *Fas* agrees with the finding that frequency of CC3[+] apoptotic kidney cells was not reduced (Supplementary Fig. 3b and Fig. 2f). As we found substantially increased expression of the negative proliferation regulators *p21/Cdkn1a* and *Btg2*, we assessed proliferation in the kidney using Ki67 and PCNA (Supplementary Fig. 3c, d)[53]. We found that, despite the increase in *p21/Cdkn1a* and *Btg2* expression (Supplementary Fig. 3e), there was an increased frequency of proliferating cells. On the surface, the increased expression of negative regulators of proliferation conflicts with the observed increase in the frequency of Ki67[+] cells. It is plausible that this discrepancy is due to the induction of proliferation needed for tissue repair to replace the cells lost by apoptosis. As the transcriptional data represents the average across cells, we would be unable to detect a population of undamaged cells undergoing proliferation to maintain tissue homoeostasis. Finally, we assessed the consequence of p53 loss on kidney morphology and function and found that neither of them was altered, with serum creatinine and urea concentrations similar to *Ercc1* single mutants (Fig. 3c–e). Together, these data show that p53 does not drive apoptosis or tissue dysfunction in the kidney of *Ercc1*[-/-] mice.

It has previously been reported that *Ercc1*-deficient mice accumulate p53 in the brain and have a range of neurodegenerative changes[7,8]. Indeed, p53 has been implicated in the neurodegenerative changes associated with DNA damage, aging and to mediate neuronal degeneration[49,50,54–57]. We therefore tested if the activation of p53 was responsible for the histophatological changes previously reported in *Ercc1*[-/-] mice. We found no difference in mass or gross morphology between *Ercc1*[-/-] and *Ercc1*[-/-] *Trp53*[-/-] brains (Supplementary Fig. 4a, b).

We found an increase in the frequency of senescent (p21[+]) cells in the brain of both *Ercc1*[-/-] and *Ercc1*[-/-] *Trp53*[-/-] mice (Supplementary Fig. 4c). We then assessed the induction of both microgliosis and astrocytosis by measuring the frequency of MAC2[+] and GFAP[+] cells, respectively. We found that both were induced in *Ercc1*[-/-] mice when compared to wildtype, however, this increase was not suppressed by the loss of p53 (Fig. 3f, g). We went on to measure inflammatory markers previously observed in *Ercc1*[-/-] mice by RT-qPCR and found that they remained induced even upon p53 loss (Supplementary Fig. 4d)[58]. Together these data show that loss of p53 is not sufficient to prevent pathological changes in the brain of *Ercc1*-deficient mice.

Another hallmark of ERCC1 deficiency is infertility and germ cell failure[11,59]. We have previously characterised the infertility defect in *Ercc1*[-/-] mice and found that *Ercc1* acts together with the Fanconi Anaemia DNA repair pathway to safeguard pre-meiotic germ cell development[11]. We found that in the absence of *Ercc1*, primordial germ cell (PGC) development is compromised by embryonic day 12.5 of development[11]. We also found that mutant PGCs accumulate pSer15-p53[+] and undergo apoptosis[11]. Moreover, p53 has been shown to drive germ cell loss in the absence of other DNA repair pathways[60]. We therefore hypothesised that loss of p53 would also rescue the germ cell defect in *Ercc1*[-/-] mice. However, we found that inactivation of p53 did not alter the reduction in either testicular or ovarian mass (Fig. 4a, b). The testis was devoid of germ cells carrying Sertoli cell-only (SCO) tubules (Fig. 4d), and the ovary had a near complete lack of follicles in both *Ercc1*[-/-] and *Ercc1*[-/-] *Trp53*[-/-] mice postnatally (Fig. 4e). However, it was in the embryonic germ cells that we had previously detected pSer15-p53[+]. We therefore quantified the frequency of PGCs at E12.5 by flow cytometry in embryos carrying the GOF18-GFP reporter and found that there was no difference between *Ercc1*[-/-] and *Ercc1*[-/-] *Trp53*[-/-] embryos (Fig. 4f)[61]. Taken together, these data show that, whilst p53 is induced in the kidney, brain and PGCs, it makes a minimal contribution to pathogenesis in those organs.

## p53 drives apoptosis and loss of *Ercc1*[-/-] haematopoietic stem cells (HSCs)

We observed a significant increase in the proportion of bone marrow cells entering p53-dependent apoptosis in the absence of ERCC1 (Fig. 2f). The bone marrow dysfunction of both humans and mice deficient in ERCC1 resembles that observed in aging and is thought to be due to reduced haematopoietic reserves. We therefore wanted to assess the role of p53 in regulating the HSC compartment, as these cells are ultimately responsible for maintaining haematopoietic homoeostasis. We assessed the transcriptomes of FACS-purified HSCs (lineage[-] c-Kit[+] Sca-1[+] CD48[-] CD150[+]) from *Ercc1*[-/-] *Trp53*[-/-] and control mice by RNA-seq[62]. We found the induction of p53 targets, including pro-apoptotic factors (*Bax*, *Fas*, among others) in *Ercc1*[-/-] HSCs that were suppressed when p53 was ablated (Supplementary Fig. 5a, b). We validated these results by performing RT-qPCR on a panel of p53-transcriptional targets that regulate different cellular responses to DNA damage (Fig. 5a). We found that whilst there was no significant increase in either of the proliferation regulators *p21/Cdkn1a* or *Btg2*, there was an induction of both apoptotic factors *Bax* and *Fas* in *Ercc1*[-/-] HSCs as expected. Remarkably, we found that, when p53 was lost, the transcriptional activation of both *Bax* and *Fas* was suppressed. These data show, in agreement with the RNA-seq and the total bone marrow data, that p53 drives an apoptotic response in the HSC compartment of *Ercc1*[-/-] mice (Figs. 1c, 2f). A key feature of the reduced haematopoietic reserve in the absence of ERCC1 is a reduced frequency of HSCs. Hence, we then set out to ask if the increased p53-dependent apoptosis in *Ercc1*[-/-] HSCs led to the observed reduction in frequency. We quantified the frequency of HSCs and HSPCs by flow cytometry and found an ~10-fold reduction in *Ercc1*[-/-]. Remarkably, this was completely rescued in *Ercc1*[-/-] *Trp53*[-/-] mice (Fig. 5b–d). These data demonstrate that transcriptional activation of apoptosis and HSC loss in *Ercc1*[-/-] mice is dependent upon p53.

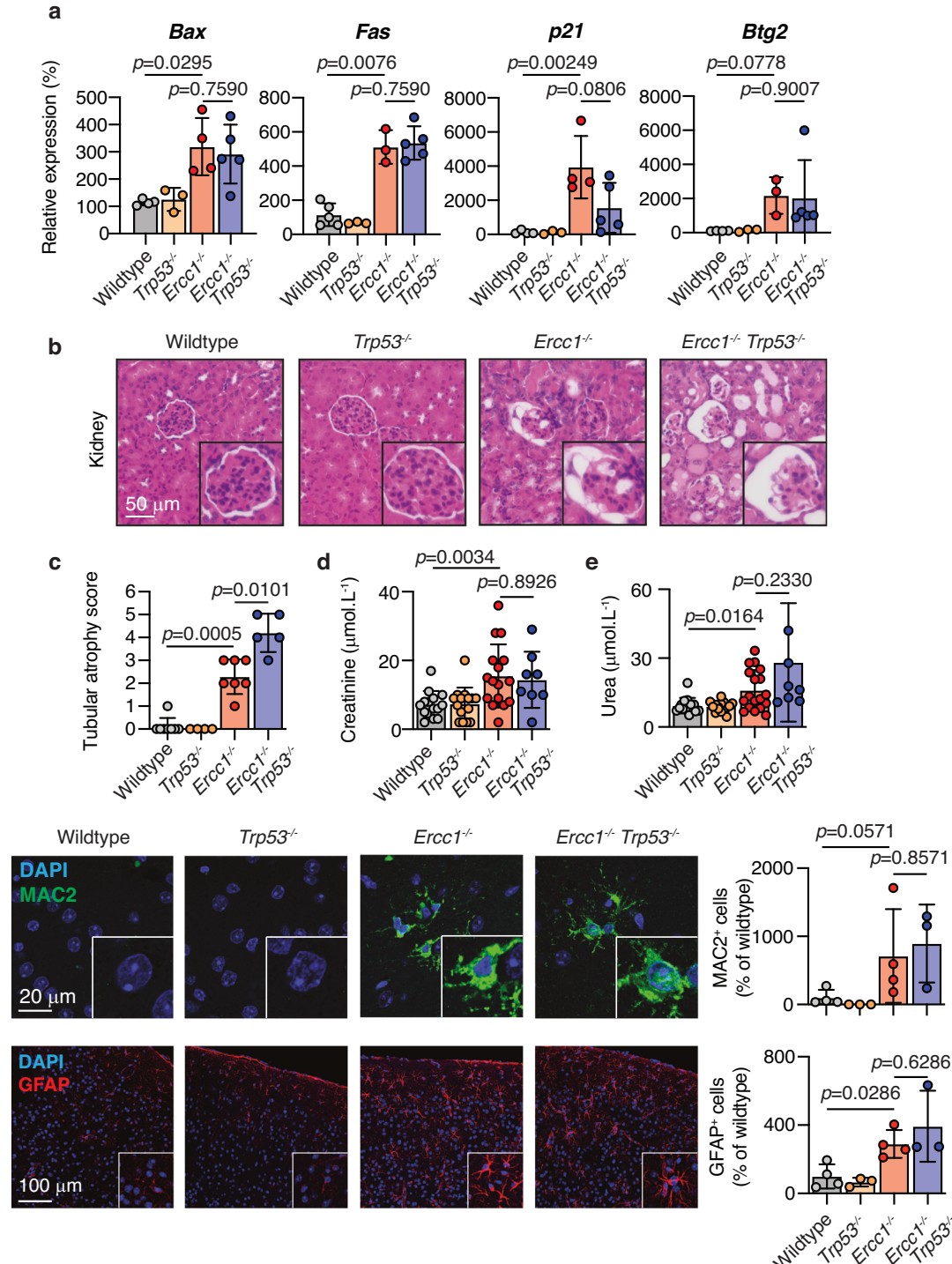

**Fig. 3 | *Ercc1⁻/⁻* kidney and neurological dysfunction are independent of p53.**
**a** Quantitative RT-PCR expression analysis of the p53 target genes (*Bax, Fas, p21* and *Btg2*) in the kidney of *Ercc1⁻/⁻Trp53⁻/⁻* and control mice (*p* values calculated by two-tailed Mann–Whitney *U*-test; data were mean ± s.d.; each dot represents an independent mouse; Bax: *n* = 4, 3, 4, 5; Fas: *n* = 5, 3, 3, 5; p21: *n* = 4, 3, 4, 5; Btg2: *n* = 4, 3, 3, 5 right to left) **b** H&E-stained sections of kidney from 6-to-8-week-old *Ercc1⁻/⁻Trp53⁻/⁻* and control mice. **c** Pathology assessment of tubular atrophy in the kidney of 6-to-8-week-old *Ercc1⁻/⁻Trp53⁻/⁻* and control mice. Scores range from 0 (absent) to 6 (substantial) (*p* value calculated by two-tailed Mann–Whitney *U*-test, data were mean ± s.d.; *n* = 8, 4, 7 and 5 independent mice, left to right). **d** Terminal blood serum creatinine levels from *Ercc1⁻/⁻Trp53⁻/⁻* and control mice (*p* value calculated by two-tailed Mann–Whitney *U*-test; *n* = 10, 8, 9 and 5

independent mice, left to right). **e** Terminal blood serum urea levels from *Ercc1⁻/⁻ Trp53⁻/⁻* and control mice (*p* value calculated by two-tailed Mann–Whitney *U*-test, data were mean ± s.d.; *n* = 10, 8, 9 and 5 independent mice, left to right). **f** Representative immunofluorescence of MAC2 staining in the cerebral cortex of *Ercc1⁻/⁻Trp53⁻/⁻* and control mice and quantification of the frequency of MAC2⁺ cells (*p* value calculated by two-tailed Mann–Whitney *U*-test, data were mean ± s.d.; *n* = 4, 3 4, and 3 independent mice, left to right). **g** Representative immunofluorescence of GFAP staining in the cerebrum of *Ercc1⁻/⁻Trp53⁻/⁻* and control mice and the frequency of GFAP⁺ cells (*p* values calculated by two-tailed Mann–Whitney *U*-test, data were mean ± s.d.; *n* = 4 wildtype, 3 *Trp53⁻/⁻*, 4 *Ercc1⁻/⁻* and 3 *Ercc1⁻/⁻Trp53⁻/⁻*). Source data are provided as a Source data file.

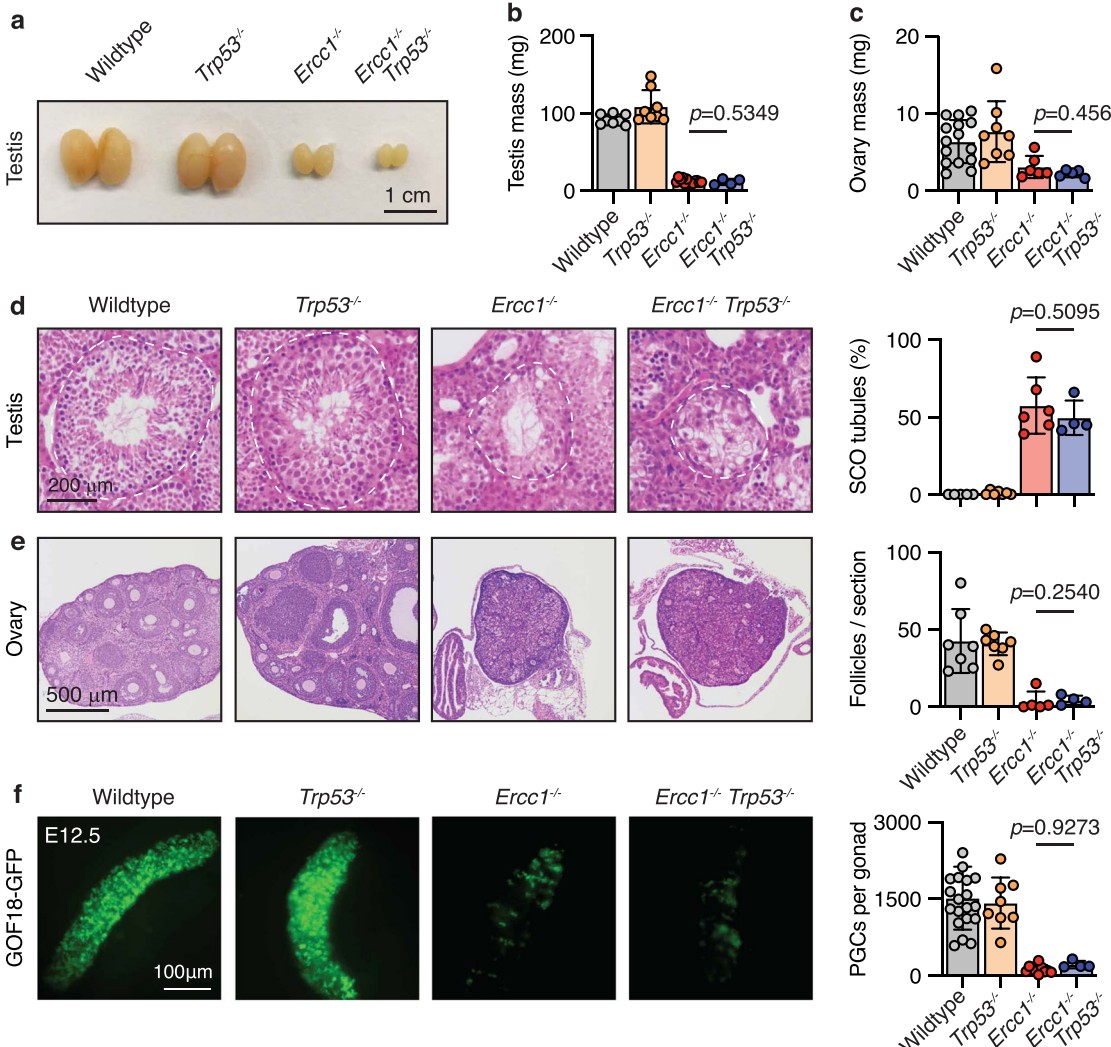

**Fig. 4 | Germ cell failure occurs independently of p53 in *Ercc1*⁻/⁻ mice. a** Representative images of testes from age-matched *Ercc1*⁻/⁻*Trp53*⁻/⁻ and control mice and **b** quantification testis mass (*p* value calculated by two-tailed Mann−Whitney *U*-test, data were mean ± s.d.; *n* = 6, 8, 10 and 4 independent mice, left to right). **c** Quantification of ovary mass from age-matched *Ercc1*⁻/⁻*Trp53*⁻/⁻ and control mice (*p* value calculated by two-tailed Mann−Whitney *U*-test, data were mean ± s.d.; *n* = 15, 8, 6 and 5 independent mice, left to right). **d** H&E-stained sections of testes from 6-to-8-week-old *Ercc1*⁻/⁻*Trp53*⁻/⁻ and control mice and the frequency of Sertoli cell-only (SCO) tubules per testis (*p* value calculated by two-tailed Mann−Whitney

*U*-test, data were mean ± s.d.; *n* = 5, 7, 6 and 4 independent mice, left to right). **e** H&E-stained sections of ovaries from 6-to-8-week-old *Ercc1*⁻/⁻ and control mice and quantification of follicles per section (*p* value calculated by two-tailed Mann−Whitney *U*-test, data were mean ± s.d.; *n* = 7, 7, 5 and 4 independent mice, left to right). **f** Representative images of GOF18-GFP fluorescence in E12.5 gonads of *Ercc1*⁻/⁻*Trp53*⁻/⁻ and control embryos and quantification of primordial germ cells (PGCs:SSEA1⁺ GOF18-GFP⁺) by flow cytometry (*p* value calculated by two-tailed Mann−Whitney *U*-test, data were mean ± s.d.; *n* = 18, 16, 4 and 7 independent embryos, left to right). Source data are provided as a Source data file.

## p53 limits liver polyploidisation in the absence of *Ercc1*

The shortened lifespan of mice lacking *Ercc1* is primarily attributed to liver failure[4,15,35]. At the cellular level, hepatocytes deficient in *Ercc1* undergo extensive polyploidization, possibly resulting from endoreduplication or a failure in cytokinesis[63,64]. p53 is a known regulator of whole genome duplication and liver ploidy; however, conflicting reports exist implicating it in both suppressing and promoting this process[32–34,31]. Moreover, previous studies have indicated the influence of p53 in hepatocyte proliferation and binucleation in *Ercc1*-deficient mice[36,37]. Therefore, we aimed to investigate: (i) whether p53 promotes or restricts liver polyploidization in *Ercc1*⁻/⁻ mice and (ii) the mechanism by which p53 regulates this process.

We first assessed the impact of p53 loss on the liver function of *Ercc1*-deficient mice. We found that there were elevated serum concentrations of ALT and AST−markers of liver damage−in the absence of *Ercc1* and that this was not affected by the loss of p53 (Supplementary Fig. 6a, b). Furthermore, when we assessed liver synthetic

function, there was a reduction in the serum concentration of albumin in the absence of *Ercc1*, which remained reduced in the *Ercc1*⁻/⁻*Trp53*⁻/⁻ mice (Supplementary Fig. 6c). These data show that loss of p53 does not subvert liver dysfunction of *Ercc1*⁻/⁻ mice, which is consistent with their dramatically reduced lifespan (Fig. 2e).

When we microscopically examined *Ercc1*⁻/⁻ livers, we observed a significant increase in the frequency of hepatocytes with enlarged and deformed nuclei consistent with previous reports (Supplementary Fig. 6d). The loss of *p53* greatly exacerbated the morphological changes observed in *Ercc1*-deficient mice (Supplementary Fig. 6d). We isolated nuclei from *Ercc1*⁻/⁻*Trp53*⁻/⁻ livers and isogenic controls and assessed their DNA content by flow cytometry[65]. Firstly, we found that there was an increase in the frequency of polyploid cells in the absence of *Ercc1* (Fig. 6a, b and Supplementary Fig. 6e). Not only did we observe an increase in the frequency of polyploid hepatocytes in *Ercc1*⁻/⁻*Trp53*⁻/⁻ mice, but we found they had undergone more extensive polyploidization, with a significant 81.6-fold increase in the

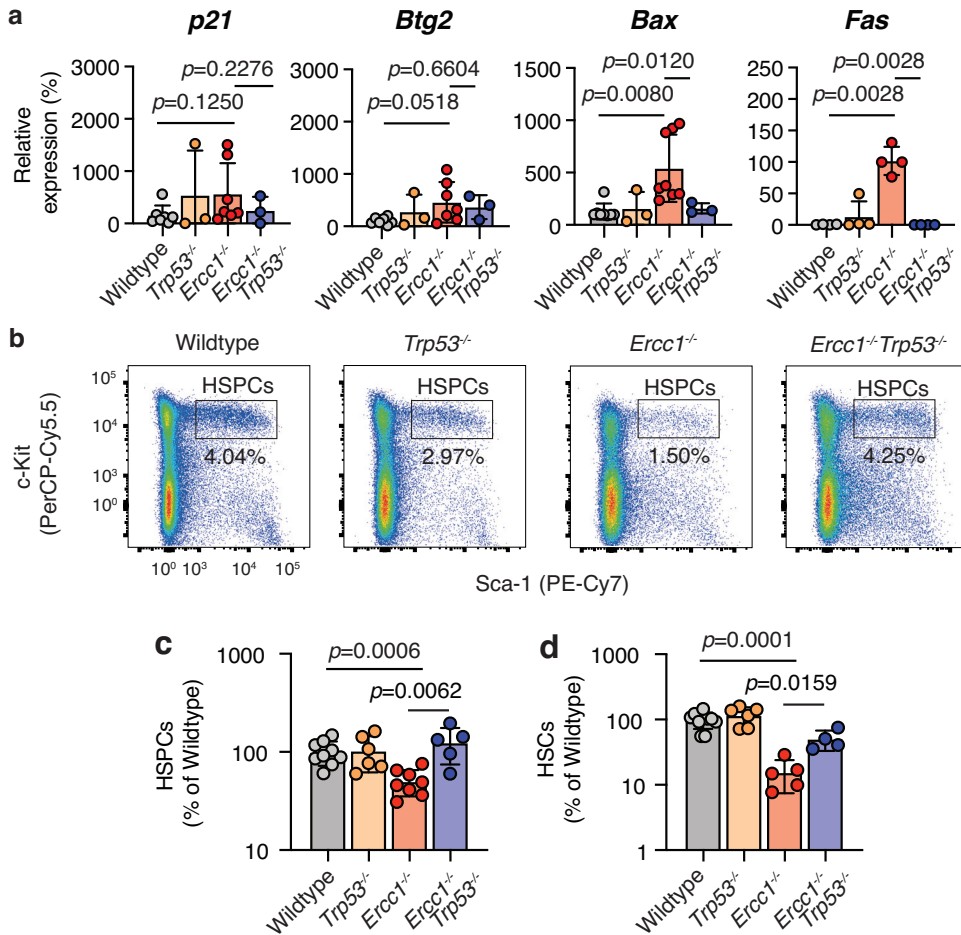

**Fig. 5 | p53 drives apoptosis and loss of *Ercc1⁻/⁻* HSCs. a** Quantitative RT-PCR expression analysis of the p53 target genes (*Bax, Fas, p21* and *Btg2*) in the HSCs of *Ercc1⁻/⁻Trp53⁻/⁻* and control mice (*p* values calculated by two-tailed Mann–Whitney *U*-test; data were mean ± s.d.; each dot represents an independent mouse. *p21*: *n* = 7, 3, 7, 3; *Btg2*: *n* = 7, 3, 7, 5; *Bax*: *n* = 8, 3, 8, 3; *Fas*: *n* = 4, 4, 4, 4; right to left **b** Representative flow cytometry plots of haematopoietic stem and progenitor cells (HSPCs, lineage⁻ c-Kit⁺ Sca-1⁺) from 6-to-8-week-old adult *Ercc1⁻/⁻Trp53⁻/⁻* and control mice. **c** Quantification of HSPCs by flow cytometry from the bone marrow of

adult *Ercc1⁻/⁻Trp53⁻/⁻* and control mice (*p* value calculated by two-tailed Mann–Whitney *U*-test, data were mean ± s.d.; *n* = 9, 6, 8 and 5 independent mice, left to right). **d** Quantification of HSCs (lineage⁻ c-Kit⁺ Sca-1⁺ CD41⁻CD48⁻ CD150⁺) by flow cytometry from the bone marrow of adult *Ercc1⁻/⁻Trp53⁻/⁻* and control mice (*p* value calculated by two-tailed Mann–Whitney *U*-test, data were mean ± s.d.; *n* = 9, 6, 5 and 4 independent mice, left to right). Source data are provided as a Source data file.

frequency of cells reaching 64n (Fig. 6a, b). These data show that the hepatocyte nuclear enlargement is due to polyploidisation and that p53 acts to limit polyploidisation in the absence of *Ercc1*.

We next asked if the role of p53 in limiting polyploidisation was specific to the liver or generalisable to other tissues from *Ercc1⁻/⁻* mice. We, therefore, measured the nuclear area, as a surrogate marker of polyploidisation, in cells from the liver, kidney, bone marrow and cerebral cortex of *Ercc1⁻/⁻* mice (Supplementary Fig. 7a). We found that only the liver exhibited a significant increase in nuclear area in the absence of *Ercc1*, whilst it led to a negligible change in the kidney, bone marrow and brain when compared to controls (Supplementary Fig. 7a, b). Furthermore, the loss of p53 only led to an increase in the nuclear area of *Ercc1⁻/⁻* hepatocytes (Supplementary Fig. 7a, b). Together, these data show that the liver uniquely undergoes polyploidisation in the absence of *Ercc1* and that this is restrained by p53.

### p16 is a failsafe proliferation brake in the absence of the p53-p21 axis

We next set out to investigate why the loss of p53 should lead to increased polyploidisation in the liver (Fig. 6a, b). A canonical function of p53 is the activation of apoptosis; however, we did not detect an

increased frequency of apoptotic hepatocytes (CC3⁺) nor transcriptional activation of apoptotic factors in *Ercc1⁻/⁻* livers (Figs. 1c, 2f and Supplementary Fig. 2a). However, despite not detecting evidence of apoptosis in hepatocytes (CC3⁺ or a transcriptional signal) we cannot exclude a role for apoptosis (e.g. at an earlier developmental stage). We also considered that the polyploidisation could be driven by an increased burden of DNA damage in *Ercc1⁻/⁻Trp53⁻/⁻* compared to *Ercc1⁻/⁻* mice. We found that whilst *Ercc1⁻/⁻* hepatocytes had a greater burden of DNA damage than wildtype, this was no different in *Ercc1⁻/⁻Trp53⁻/⁻* mice, suggesting that the absence of p53 does not lead to the accumulation of more DNA breaks (Fig. 6c, d and Supplementary Fig. 7c).

Finally, we examined whether the involvement of p53 in proliferation regulation provides insight into its role in controlling ploidy. In order for polyploidisation to occur, hepatocytes must either undergo iterative genome duplications or nuclear fusion events[64]. We hypothesised that a p53-dependent cell-cycle arrest mechanism was lost in *Ercc1⁻/⁻Trp53⁻/⁻* hepatocytes, leading to uncontrolled cell-cycle progression. To test this, we stained the livers of young 3-to-6-week-old mice for the marker of proliferation Ki67 and observed a significant increase in the frequency of Ki67⁺ hepatocytes in *Ercc1⁻/⁻Trp53⁻/⁻* compared to *Ercc1⁻/⁻* mice (Fig. 6e, f). Furthermore, we found that, in

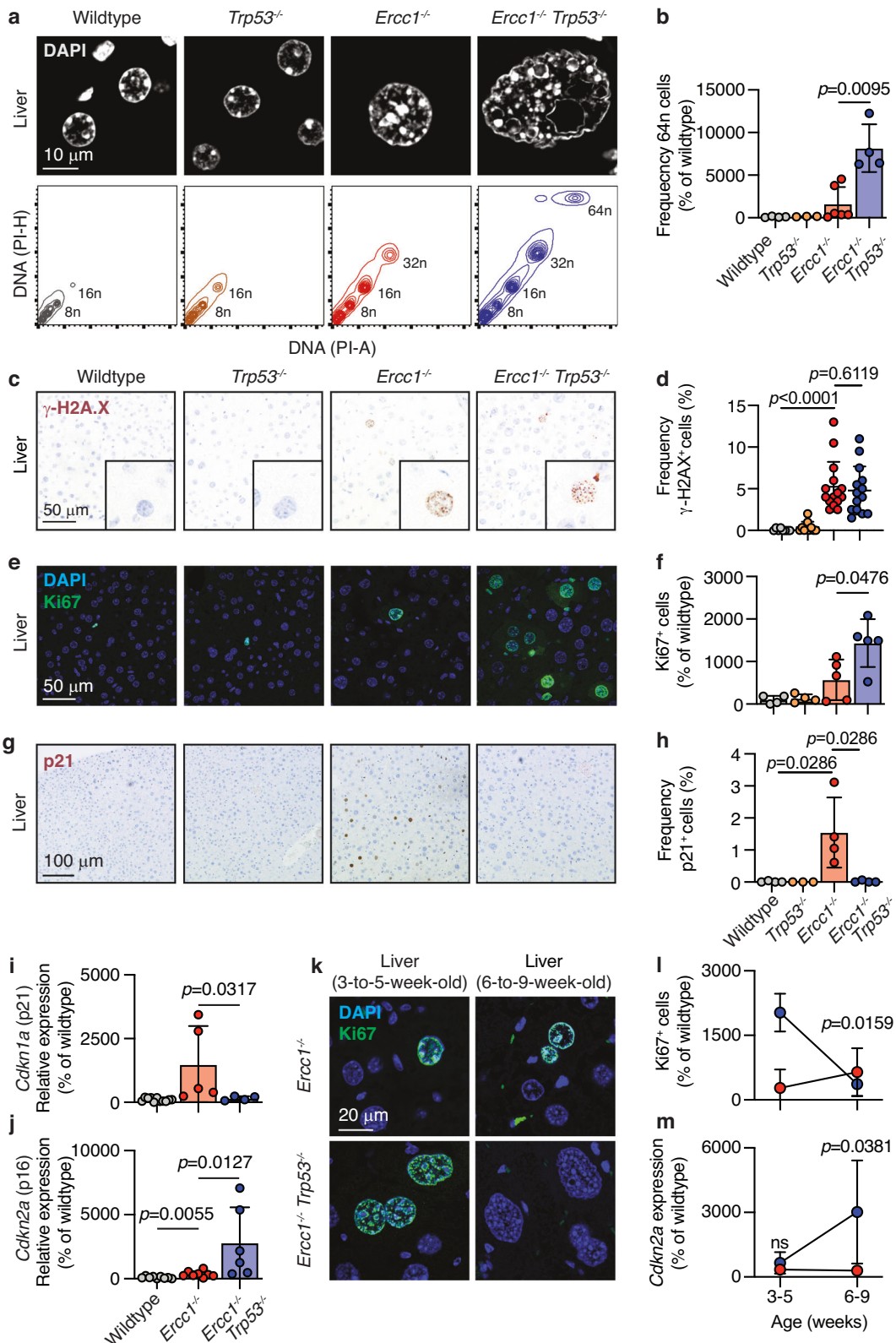

the absence of *Ercc1*, there was a dramatic induction of the negative cell-cycle regulator *Cdkn1a/p21*; however, this is entirely suppressed when p53 is ablated (Fig. 6g–i)[66–68]. It is likely that loss of the p53-p21 axis allows further hepatocyte proliferation enabling polyploidisation. Therefore, the massive polyploidisation in *Ercc1*-deficiency is likely the result of endoreduplication, the process of genome replication without mitosis.

Polyploidisation is often observed upon liver aging; indeed, *Ercc1* deficiency has been proposed as a progeroid model. However, liver polyploidisation can occur either by failed cytokinesis, often associated with age-dependent polyploidisation or by the process of endoreduplication more associated with damage and regeneration[69]. We performed bulk RNA sequencing on purified hepatocytes from *Ercc1*[−/−] and control mice (Supplementary Fig. 7d). We first confirmed

**Fig. 6 | p16 is a failsafe proliferation brake in the absence of the p53-p21 axis.**
**a** Top−Representative immunofluorescence of DAPI staining in the liver and quantification of hepatocyte nuclear area (p values calculated by two-tailed Mann−Whitney U-test, data were median and interquartile range; n = 150 hepatocytes per genotype, 50 per mouse). Bottom− Representative flow cytometry plots of DNA content in the nuclei of liver cells of 6-to-8-week-old mice. **b** Quantification of the frequency of 64n cells (p value calculated by two-tailed Mann−Whitney U-test, data were mean ± s.d.; n = 4, 3, 6 and 4 independent mice, left to right). **c** Representative immunofluorescence of γ-H2A.X staining in the liver. **d** Quantification of γ-H2A.X$^+$ cells (p value calculated by two-tailed Mann−Whitney U-test, data were mean ± s.d.; each dot corresponds to individual mice; n = 8, 10, 16, 14 left to right). **e** Representative immunofluorescence of Ki67 staining in the liver. **f** Quantification of Ki67$^+$ cells (p value calculated by two-tailed Mann−Whitney U-test, data were mean ± s.d.; n = 4, 4, 5 and 5 independent mice, left to right).

**g** Representative immunofluorescence of p21 staining in the liver. **h** Quantification of p21$^+$ cells (p value calculated by two-tailed Mann−Whitney U-test, data were mean ± s.d.; each point represents one independent mouse, n = 4, 3, 4, 4 left to right). **i, j** Quantitative RT-PCR expression analysis of Cdkn1a (top) and Cdkn2a (bottom) in the liver (p values calculated by two-tailed Mann−Whitney U-test, data were mean ± s.d.; n = Cdkn1a (11, 5 and 4) and Cdkn2a (9, 8 and 6), mice, left to right). **k, l** Representative immunofluorescence of Ki67 staining in the liver of young (3-to-5-week-old) and older (6-to-9-week-old) mice and quantification of the frequency of Ki67$^+$ hepatocytes (p values calculated by two-tailed Mann−Whitney U-test, data were mean ± s.d.; n = 4 independent mice per genotype and timepoint). **m** Quantitative RT-PCR expression analysis of p16/Cdkn2a in the liver of young (3-to-5-week-old) and older (6-to-9-week-old) mice (p values calculated by two-tailed Mann−Whitney U-test, data were mean ± s.d.; n = 4 independent mice per genotype and timepoint). Source data are provided as a Source data file.

that, in the absence of Ercc1, p21/Cdkn1a was induced and that this was suppressed when p53 was ablated (Supplementary Fig. 7e). We next assessed the expression of a 'liver aging signature' defined in the Mouse Aging Cell Atlas (Tabula Muris Senis)[70]. Interestingly, we observed a strong concordance between the hepatocytes of young 6-to-8-week-old Ercc1$^{-/-}$ mice and the aging signature (Supplementary Fig. 8a). We then assessed the transcriptional changes when p53 was ablated and found that the aging signature persisted (Supplementary Fig. 8a). One hallmark of aging is dysregulated nutrient signalling as the growth hormone (GH)/insulin-like growth factor (IGF) somatotrophic axis is downregulated both in natural aging and in models of premature aging[71–74]. It has previously been shown that Ercc1-deficient mice suppress the GH/IGF somatotrophic axis and it is hypothesised that this attenuation diverts resources from growth towards a pro-survival stress response to cope with systemic damage[4]. However, it remains unclear how the loss of ERCC1 leads to attenuated endocrine signalling. We asked if p53 signalling contributes to the dysregulation of GH/IGF signalling. Consistent with previous studies, our analysis of the differentially expressed genes in Ercc1$^{-/-}$ hepatocytes compared to wildtype controls revealed altered expression of the REACTOME pathway "MMU-381426 - regulation of IGF transport and uptake by IGF binding proteins (IGFBPs)" (p = 4.68 × 10$^{-15}$, rank 3rd). Strikingly, when we compared the transcriptome of Ercc1$^{-/-}$Trp53$^{-/-}$ hepatocytes with Ercc1$^{-/-}$ controls the most enriched REACTOME pathway was MMU-381426 (p = 3.67 × 10$^{-7}$, rank 1st), demonstrating that dysregulation of the IGF signalling pathway persists even in the absence of p53 (Supplementary Fig. 8b, c). We also observed transcriptional inflammatory and immune responses in Ercc1$^{-/-}$Trp53$^{-/-}$ livers which are also characteristic of aged tissues (Supplementary Fig. 8b)[71]. Together, these led us to conclude that Ercc1$^{-/-}$Trp53$^{-/-}$ liver had multiple transcriptional changes reminiscent of premature aging.

Furthermore, one of the best-characterised hallmarks of aging is cellular senescence, which is not only a marker of increased age but also of tissue dysfunction[13,70,71,75,76]. We therefore sought to determine if the p53 response regulates senescence in the liver of Ercc1$^{-/-}$ mice. To do this, we quantified the expression of the well-characterised marker of senescence and cell-cycle regulator, p16/Cdkn2a. Surprisingly, we found that the liver of Ercc1$^{-/-}$Trp53$^{-/-}$ mice had a significant induction in the expression of p16/Cdkn2a compared to Ercc1$^{-/-}$ controls (Fig. 6j). This appeared contradictory to our previous results showing that the loss of the p53-p21 led to increased proliferation of Ercc1$^{-/-}$Trp53$^{-/-}$ hepatocytes (Fig. 6e, f). To reconcile this, we compared p16/Cdkn2a expression and proliferation in the liver at different time points. When we compared the frequency of proliferating hepatocytes in young (3-to-5-week-old) and older (6-to-9-week-old) Ercc1$^{-/-}$Trp53$^{-/-}$ mice, we observed a significant reduction in the frequency of Ki67$^+$ cells with age (Fig. 6k, l). We confirmed this result by staining the liver of old (6-to-9-week-old) Ercc1$^{-/-}$Trp53$^{-/-}$ mice with the alternative proliferation marker PCNA (Supplementary Fig. 8d). Concomitantly, we observed a significant increase in the relative expression of p16/Cdkn2a in the liver

of Ercc1$^{-/-}$Trp53$^{-/-}$ mice (Fig. 6m). These data show that, in the absence of the p53-p21 axis, p16/Cdkn2a is induced, and proliferation is suppressed, a defining feature of cellular senescence. In the absence of the p53-p21 axis, we see engagement of p16 and arrested proliferation. Given the documented role of p16 in cell-cycle regulation, this may represent a failsafe mechanism to prevent proliferation and endoreduplication.

## The p53-axis protects the liver from chemotherapy-induced polyploidisation

The XPF-ERCC1 endonuclease acts to counteract several forms of DNA damage, including helix-distorting adducts, intra- and inter-strand crosslinks[3,77–79]. However, it is unclear which of these lesions could drive the liver phenotype. We hypothesised that inter-strand crosslinks could likely contribute to liver polyploidisation as mice deficient in the ability to repair ICLs (Fan1$^{-/-}$, Adh5$^{-/-}$Fancd2$^{-/-}$) also exhibit liver polyploidisation[65,80]. We turned our attention to platinum-based DNA inter-strand crosslinking agents, as these are also frequently used in chemotherapy, including the treatment of hepatocellular carcinoma[81,82]. We exposed wildtype mice to cisplatin and observed a significant increase in the frequency of cells with >5 nuclear γ-H2A.X foci in the liver (Supplementary Fig. 9a, b). Moreover, the treatment led to the accumulation of pSer15-TP53 in the liver (Supplementary Fig. 9a, b). These responses are similar in magnitude to what was observed spontaneously in Ercc1$^{-/-}$ mice (Fig. 1a, b). Therefore, we next wanted to ask if this induction of p53 acted to restrain liver polyploidisation in the face of exogenous challenge with DNA inter-strand crosslinking agents. We found that chronic exposure to cisplatin-induced a significant increase in the frequency of polyploid (>16n) cells in the liver of Trp53$^{-/-}$ mice, which was not observed in p53-proficient mice (Supplementary Fig. 9c, d). These data demonstrate that DNA crosslinking agents can induce liver polyploidisation and that the same p53-dependent mechanism acts to restrain polyploidisation following exposure to chemotherapy.

## Discussion
The role of p53 as a regulator of the cellular response to DNA damage has long been recognised[20,41]. Remarkably, the loss of p53 rescues cellular hypersensitivity of both wildtype and DNA repair-deficient cells to a wide array of DNA damaging agents in vitro[23,46,83–85]. Moreover, p53 is known to contribute to the phenotype of animal models of DNA repair deficiency syndromes and can suppress phenotypes, such as lethality, developmental defects and blood stem cell loss[43–50]. The counter-side of p53 loss is an increased burden of DNA damage, mutation and neoplastic transformation[43–45,49].

Here we find that the loss of ERCC1 results in the induction of the DNA damage marker γ-H2A.X and activation of p53 due to endogenous DNA damage with different transcriptional and pathological consequences. We show that there is induction of p53-dependent apoptosis in bone marrow with ablation of p53 restoring HSC number. In

contrast, loss of p53 did not rescue infertility, neurodegenerative changes, or kidney dysfunction. On the other hand, p53 ablation resulted in further polyploidization of the liver of *Ercc1*⁻/⁻ mice. Therefore, p53 plays a critical role in regulating tissue-specific outcomes of DNA repair deficiency.

The observation that p53 loss rescues the HSC defect clearly demonstrates that, at least in the blood stem cell pool, the phenotype is driven by the p53 response rather than HSC loss as a direct consequence of DNA damage. It has previously been shown that p53 loss protects HSCs from both endogenous and exogenous DNA damage[26,45,46,86]. Previous work has also shown that the Fanconi anaemia (FA) HSC defect in *Fancd2*−/− mice is rescued by p53 loss, but not p21 loss[45,87]. This, together with the apoptotic response data shown here, indicates that the reduction in *Ercc1*/- HSC number is largely mediated by p53-dependent apoptosis and likely due to the role of *Ercc1* in the FA pathway.

In contrast, loss of p53 does not rescue the infertility of *Ercc1*/- mice. We have previously shown that *Ercc1*-deficient PGCs have persistent DNA damage, p53 activation and undergo apoptosis[11]. However, we now show that p53 ablation does not restore germline development. This is surprising in light of the ability of p53 to rescue the HSC defect but also as p53 loss can partly restore the PGC number in the absence of *Fancm* (a different DNA repair factor)[60]. These data raise the interesting possibility that PGCs may employ a different p53-independent quality control mechanism to preserve genome stability akin to what is observed in oocytes[88]. Recent work has identified the DREAM complex as a repressor of DNA repair in somatic tissues but not in the germ line[14].

Interestingly, it is thought that the activation of *p53* in the brain of *Ercc1*-deficient mice may contribute to pathological changes, but we find that the loss of p53 does not rescue the neurodegenerative changes observed in *Ercc1*⁻/⁻ mice[7,8]. Ablation of p53 can attenuate neuron loss in a range of different DNA repair-deficient mice[50,55–57,89], suggesting the origin of the neurological changes observed in *Ercc1*⁻/⁻ mice is different. It has been shown that there is a cell-intrinsic requirement for *Ercc1* in neurons[8]. However, there is also evidence of uraemic encephalopathy due to renal dysfunction in *Ercc1*⁻/⁻ mice[90]. Therefore, it is possible that p53 has both a cell-intrinsic role in the neuropathological changes but that the renal dysfunction observed in *Ercc1*⁻/⁻ *Trp53*⁻/⁻ mice may also contribute to the neurodegeneration.

Moreover, we found that p53 loss is unable to rescue renal function. Despite activation of the p53-p21 axis, we observed an increased frequency of proliferating (Ki67⁺ or PCNA⁺) cells. Even in the absence of p53, we detect a significant increase in the frequency of p21⁺ cells. p21 activation is often downstream of p53 however, p53-independent activation of p21 is well documented[91,92]. Determining the molecular basis of p21 activation in *Ercc1*⁻/⁻ *Trp53*⁻/⁻ kidneys will be an important area of future study. We also found activation of apoptotic factors and an increase in apoptotic cells that persisted even when p53 was lost. This suggests that p53-independent apoptosis makes an important contribution to tissue homoeostasis in the kidney. p53-independent apoptosis in response to DNA damage has previously been described, and it will be interesting to elucidate this mechanism in future work[85,93]. Recent work has shown that senescent cells may undergo mitochondrial outer membrane permeabilization, which may contribute to the observed activation of senescence and apoptosis markers in *Ercc1*/- kidneys[94]. Indeed, the kidney is a complex organ made up of many different cell types, with different embryonic origins and replicative potentials, and p53 has been shown to play different roles in different sections of the nephron[95]. It will be an area of future research to uncover how different sections of the nephron respond to DNA damage and what factors and checkpoints are involved in this signalling.

*Ercc1*⁻/⁻ mice are born with diploid hepatocytes that become progressively polyploid. There are conflicting reports on the role of p53 in the regulation of ploidy in response to DNA damage[31–34]. We find that *Ercc1*⁻/⁻ hepatocytes undergo cell-cycle arrest when cells acquire a chromosome complement of 8-16n. This arrest is associated with the p53-dependent expression of *p21/Cdkn1a*. Inactivation of the p53 pathway leads to the loss of both p21 expression and cell-cycle regulation, leading to continued endoreduplication and chromosome complements >64n. This is associated with the upregulation of an alternate cell-cycle regulator, *p16/Cdkn2a*, suggesting that p16-dependent cell-cycle arrest may act as a failsafe to restrain further hepatocyte polyploidisation in the absence of the p53-p21 axis. Previous studies have shown that p53 is activated in the liver of *Ercc1*⁻/⁻ mice, but that loss of p53 did not rescue the liver dysfunction. However, p53 was responsible for the reduced proliferation and observed binucleation[36,37]. We add significantly to these initial findings; firstly, we find that in the absence of p53 not only do more hepatocytes become polyploid, but that the extent of polyploidisation increases. Secondly, there is a temporal aspect of the phenotype; initially, the loss of p53 leads to increased proliferation of the hepatocytes, but remarkably by 6 weeks, this increase is suppressed. Thirdly, we provide an explanation for this apparent paradox—that despite the loss of the p53-p21 cell-cycle regulator, proliferation is halted. We find that by 6 weeks, there is robust activation of *p16/Cdkn2a* coinciding with the brake to proliferation. Together, our results show that, in *Ercc1*-deficient hepatocytes, p53 is a negative regulator of polyploidisation.

It is possible that different tissues experience different burdens or classes of DNA damage. XPF-ERCC1 is involved in at least three distinct repair transactions—nucleotide excision repair (NER), FA crosslink repair and homologous recombination[78,96–99]. However, mice deficient in both NER and FA crosslink repair pathways do not recapitulate the phenotype of *Ercc1*-deficient mice suggesting a distinct role for ERCC1[38]. It is, therefore, plausible that distinct outcomes in different tissues could be due to the relative importance of these repair processes between tissues. Together, this work finds a surprising diversity of tissue-specific outcomes that are regulated by p53.

## Methods

### Mice

All animal experimentation was undertaken in this study were approved by the Medical Research Council's Laboratory of Molecular Biology animal welfare and ethical review body and the UK Home Office under the Animal (Scientific Procedures) Act 1986 (licence number PP6752216). All mice used in this study were maintained under pathogen-free conditions and housed in individually ventilated racks (GM500; Techniplast) on Lignocel FS-15 spruce bedding (IPS) and provided with environmental enrichment (chew stick, fun tunnel and Enviro-Dri nesting (LBS)). Animals were maintained at 19–23 °C with light from 07:00 to 19:00 and fed Dietex CRM pellets (SDS) ad libitum. No were wild, and no field-collected samples were used in this study. Unless otherwise stated, mice were generated on a C57BL/6 x 129S4S6/Sv F1 genetic hybrid background. Samples were collected from adult mice at various stages, as stated in the text. The investigators were blinded to the genotypes of animals throughout the study and data were acquired on the basis of identification numbers only. *Ercc1*^tm1a(KOMP)Wtsi^ (MGID: 4362172), *Trp53*^tm1Brd^ (MGID: 1857590) and GOF18-GFP (*Tg(Pou5f1-EGFP)11Ymat*) (MGID: 6148237) have been described previously.

### Cisplatin treatment of mice

Unless otherwise stated, 6-to-8-week-old wildtype or *Trp53*⁻/⁻ mice were administered cisplatin at 4 mg.kg⁻¹ via intraperitoneal injection (IP) at 10 ml.kg⁻¹ once per week for a period of 6 weeks. Subsequently mice were either culled 24 h after the last IP or mice were monitored for a period of 3 months.

## Histological analysis

Tissues were fixed in 10% neutral-buffered formalin for 24–36 h and transferred into 70% ethanol. Femurs and sternum samples were decalcified and embedded in paraffin, and 4 µm sections were cut before staining with haematoxylin and eosin using standard methods.

## Immunohistochemistry and assessment of DNA damage markers

Sections of formalin-fixed, paraffin-embedded samples were deparaffinised and rehydrated using xylene and 100–95% ethanol gradient following standard methods. Slides were equilibrated in antigen retrieval buffer (10 mM sodium citrate, pH 6.0) and boiled for 10 min. Subsequently, samples were washed in water three times for 5 min and then once in tris-buffered saline (TBS)/0.1% w/v Tween 20 for 5 min. Samples were then blocked in blocking buffer (TBS, 0.1% w/v Tween 20, 5% v/v goat serum) for 1 h at room temperature and incubated overnight at 4 °C with the following primary antibodies; Anti-phospho-Ser139-Histone H2A.X (1:1000, catalogue no. 05-636; Merk Millipore), anti-phospho- Ser15-TP53 (1:500, catalogue no. D4S1H; Cell Signalling Technology), anti-cleaved caspase-3 (1:300, catalogue no. D175; Cell Signalling Technology), anti-glial acidic filament protein (GFAP) (1:500, catalogue no. Z0334; Dako), anti-MAC2 (1:2000, catalogue no. 125402; BioLegend), anti-Ki67 (1:100, catalogue no. M3062; Spring Science), Anti-p21 (1:500, [HUGO291], ab107099, Abcam), anti-PCNA (1:1000, PCNA (PC10) Mouse mAb Cell signalling catalogue no. 2586). Slides were then washed in TBS, 0.1% w/v Tween 20 for 5 min three times and incubated at room temperature with the following secondary antibodies for one hour at room temperature; goat anti-mouse Alexa Fluor 488 (1:1000, catalogue no. A11029; Thermo Fisher Scientific) and goat anti-rabbit Alexa Fluor 594 (1:1000, catalogue no. A11037; Thermo Fisher Scientific). The slides were washed in TBS, 0.1% w/v Tween 20 for 5 min three times and stained with 0.5 µg.mL$^{-1}$ DAPI diluted in PBS for 10 min at room temperature. Slides were then mounted with Pro-Long Gold antifade reagent and coverslips were placed onto slides. Images were acquired on an LSM 780 confocal microscope (Zeiss). Image analysis was performed using NIS-Elements (Nikon) software to quantify γ-H2A.X foci per nucleus, nuclear area and the frequency of p-TP53$^+$, CC3$^+$, GFAP$^+$ and MAC2$^+$ cells. p21, γ-H2A.X and PCNA staining was performed by DAB IHC using the Leica bond platform.

## Embryo isolation and foetal gonad dissection

Foetal gonads were isolated from timed matings that were performed overnight, and females were assessed for the presence of copulation plugs the following morning. Halfway through the light cycle of the day, a copulation plug was recorded and was designated as embryonic day 0.5 (E0.5). Pregnant females were killed by cervical dislocation and exsanguination at noon on the appropriate day of gestation (E12.5), and then embryos were placed into ice-cold PBS. Embryos were dissected, and the genital ridge was placed into ice-cold PBS until further analysis.

## DNA content analysis by flow cytometry

Fresh livers and kidneys were isolated from age-matched mice and stored in ice-cold PBS on ice. Subsequently, 250 mg of tissue was passed through a 70-µm cell strainer and washed twice in buffer LA (250 mM sucrose, 5 mM MgCl$_2$, 10 mM Tris-HCl, pH 7.4). Cells were resuspended washed twice in 1 mL of buffer LB (2 M sucrose, 1 mM MgCl$_2$, 10 mM Tris-HCl, pH 7.4) and centrifuged at 16,000×$g$ at 4 °C for 30 min. Subsequently, nuclei were resuspended in 1 mL buffer LA and transferred to a fresh 15 mL Falcon tube and fixed in 70% ethanol overnight at −20 °C. Nuclei were resuspended in staining buffer (PBS supplemented with propidium iodide (40 µg.mL$^{-1}$, Sigma) and Ribonuclease A from bovine pancreas (100 µg.mL$^{-1}$, Sigma)) and incubated at room temperature for 20 min. Samples were then immediately run

on an LSRII FACS analyser (BD Pharmingen), and the data analysed with FlowJo v.10.1r5. The flow cytometry gating strategy can be found in Supplementary Fig. 11.

## Clinical biochemistry

Blood serum was harvested from 200 to 500 µL whole blood obtained via cardiac puncture into a Microvette® 500 Z-Gel tube. Serum levels of urea, albumin, aspartate aminotransferase and alanine aminotransaminase were determined using a Siemens Dimension® RxL Max integrated chemistry system. Proteinuria was assessed by running 5 µl of fresh urine into a 4–12% precast SDS-PAGE gel and staining with Coomassie. Full uncropped gel can be found in Supplementary Fig. 10.

## Quantification and isolation of HSPCs and PGCs in vivo

For PGC quantification, the genital ridge of developing embryos carrying the *GOF18-GFP* reporter were isolated as described above and placed into 150 µL trypsin solution pre-warmed to 37 °C and incubated for 10 min at 37 °C in a water bath. Subsequently, 1 µL Benzonase (99% purity, Millipore) was added and the sample was disaggregated by gentle pipetting and incubated for a further 5 min at 37 °C in a water bath. Trypsin was inactivated by adding 1 mL of PBS/5% v/v FCS, and samples were centrifuged at 1100×$g$ for 10 min. The supernatant was discarded and the cell pellet resuspended in 100 µL of anti-SSEA1 antibody conjugated to Alexa Fluor 647 (1:100, catalogue no. MC-480; BioLegend) and incubated at room temperature for 10 min. Samples were then diluted by adding 300 µL PBS/5% v/v FCS and immediately run on an ECLIPSE analyser (Sony Biotechnology), and the data analysed using FlowJo v.10.1r5 (FlowJo LLC). PGCs were defined as SSEA1$^+$ and GOF18-GFP$^+$. Flow cytometry gating strategy can be found in Supplementary Fig. 11. For HSC quantification, bone marrow cells were harvested from the femurs and tibiae of 6-to-8-week-old adult mice in staining buffer (PBS/2.5% v/v FCS) and filtered through a 70-µm cell strainer. Erythrocytes were lysed by resuspending samples in 10 mL of red cell lysis buffer (MACS Miltenyi Biotec) for 10 min at room temperature. Cells were resuspended in 1 mL of staining buffer, and the number of nucleated cells quantified by diluting 10 µL of cell suspension in 990 µL 3% acetic acid solution (StemCell Technologies) and quantified using a Vi-Cell XR cell viability counter (Beckman Coulter). Subsequently, $10 \times 10^6$ nucleated bone marrow cells were stained in 200 µL of staining solution containing the following antibodies; FITC-conjugated lineage cocktail (anti-CD4 (clone H129.12; BD Pharmingen), anti-CD3e (clone 145-2 C11, eBioscience), anti-Ly-6G/Gr-1 (clone RB6-8Cs, eBioscience), anti-CD11b/Mac-1 (clone M1/70, BD Pharmingen), anti-CD45R/B220 (clone RA3-6B2, BD Pharmingen), anti-Fcɛ R1α (clone MAR-1, eBioscience), anti-CD8a (clone 53-6/7, BD Pharmingen), anti-CD11c (clone N418, eBioscience), anti-TER-119 (clone Ter119, BD Pharmingen) and anti-CD41 (1:100 clone MWReg30, BD Pharmingen); anti-c-Kit (1:100 PerCP-Cy5.5, clone 2B8, eBioscience), anti-Sca-1 (1:100 PE-Cy7, clone D7, eBioscience), anti-CD150 (1:100 PE, clone TC15-12F12.2, eBioscience), and anti-CD48 (1:100 biotin, clone HM48.1, BioLegend). Samples were incubated with the primary antibodies for 15 min at room temperature and washed in 1 mL of staining buffer. Samples were then resuspended in 200 µL of staining buffer containing streptavidin-BV421 and incubated for 15 min at room temperature. Finally, samples were washed in 1 mL staining solution and resuspended in 400 µL of staining buffer and immediately run on an LSRII FACS analyser (BD) and the data were analysed using FlowJo v.10.1r5 (FlowJo LLC). HSPCs were defined as lineage$^-$, CD41$^-$, Sca-1$^+$, and c-Kit$^+$ and HSCs define as lineage$^-$, CD41$^-$, Sca-1$^+$, c-Kit$^+$, CD48$^-$ and CD150$^+$. The gating strategy is shown in Supplementary Fig. 11.

## Purification of hepatocytes

Hepatocytes were purified from fresh liver. The liver was harvested into ice-cold RPMI media and subsequently placed into liver digestion media (Hanks' Balanced Salt Solution (HBSS) without calcium or

magnesium, supplemented with 120 U.mL⁻¹ Type III collagenase) and incubated at 37 °C for 30 min with gentle agitation. The liver was then homogenised, passed through a 70-µm cell strainer, and the digestion media inactivated with the addition of 5 mL blocking buffer (HBSS supplemented with 5 mM EDTA and 2% v/v FBS). Samples were centrifuged and resuspended in 5 mL ice-cold red cell lysis buffer (MACS Miltenyi Biotec) and incubated for 3 min on ice. Samples were diluted with 25 mL RPMI/5% v/v FCS and passed through a 70-µm cell strainer, and centrifuged. Cell pellets were resuspended in 5 mL RPMI/5% v/v FCS and layered over a 10 mL Lymphoprep™ cushion, centrifuged at 800 × g for 25 min at 20 °C. The cell pellet containing mainly hepatocytes was transferred to a fresh 15 mL Falcon tube and washed three times in 5 mL RPMI/5% v/v FCS for 5 min. Finally, hepatocytes were resuspended in RPMI/5% v/v FCS and mixed at a 1:1 ratio with isotonic Percoll® (1.07 g.mL⁻¹) and centrifuged at 800 × g for 5 min. Subsequently, the upper fraction containing mainly hepatocytes was collected, and washed in PBS, and cell pellets were stored at −80 °C until further analysis.

### Quantitative RT-PCR and gene expression analysis
HSCs, liver and kidney cells were isolated as described earlier, and total RNA was extracted using the RNeasy Mini Kit (QIAGEN). First-strand complementary DNA synthesis was performed using the QuantiTect Reverse Transcription Kit (QIAGEN) following the manufacturer's instructions. Quantitative real-time PCR analysis for the expression of *Cdkn1a*, *Btg2*, *Mdm2*, *Gdf15* and *Bax* was performed using the following oligonucleotides using Brilliant II SYBR Green QPCR Master Mix (Agilent) using a ViiA 7 Real-Time PCR system (Thermo Fisher Scientific) at 95 °C for 10 min and 40 cycles of 95 °C for 15 s and 60 °C for 1 min. Mean thresholds were determined from three technical repeats per sample and oligonucleotide pair using standard comparative $C_T$ methods. All expression levels were normalised to the housekeeping gene *Gapdh*. The sequence of oligonucleotides is available in Supplementary Data 1.

### TUNEL assay
The TUNEL assay was performed using the Click-iT TUNEL Colorimetric IHC detection kit from Life Technologies (cat 10625), following the manufacturer's instructions, with some modifications. First, slides were incubated for 20 min in proteinase K solution and 10 min in DAB reaction mixture. The deparaffination was performed by incubations in xylene 5 min, xylene 5 min, ethanol 100% 30 s, ethanol 100% 30 s, ethanol 96% 30 s, ethanol 90% 30 s, ethanol 80% 30 s, ethanol 70% 30 s, ethanol 60% 30 s, ethanol 25% 30 s and water 5 min, twice. Finally, for counterstaining, slides were incubated in Mayer's haematoxylin for 3 min, water 1 min, ethanol 50% 30 s, ethanol 60% 30 s, ethanol 70% 30 s, ethanol 96% 30 s three times, ethanol 100% 5 min three times and xylene 3 min, twice.

### Bulk RNA sequencing
HSCs, PGCs and hepatocytes were isolated as described earlier. For HSCs and PGCs, RNA sequencing libraries were prepared using the NEBNext® Single Cell/Low Input RNA Library Prep Kit for Illumina® kit and NEBNext® Poly(A) mRNA Magnetic Isolation module kit following the manufacturer's instructions. For hepatocytes, sequencing libraries were prepared using the NEBNext® Ultra™ II RNA Library Prep Kit for Illumina® kit following the manufacturer's instructions. Subsequently, all libraries were ligated to NEBNext® Multiplex Oligos for Illumina® (Dual Index Primer Set I), and quality control was performed using a 2100 Bioanalyser High Sensitivity DNA Kit 5067-4626 (Agilent) and libraries quantified using a Qubit™ Fluorometer following the manufacturer's instructions. Finally, RNA sequencing libraries were pooled to a final concentration of 8.5 nM and quality control sequencing analysis was performed on a MiSeq system (Illumina) before being loaded onto a single S1 chip on a NovaSeq 6000 system (Illumina) and

sequenced using 150 bp-paired end sequencing reactions. Raw reads were quality-trimmed using trim_galore (v0.4.4). Trimmed reads were then mapped to the GRCm38 *Mus musculus* genome using hisat (v2-2.2.1) and differential expression analysis performed with cuffdiff (v2.2.1). Gene set enrichment analysis was performed with g:Profiler.

### Statistical analysis
Unless otherwise stated in the text, data are shown as the mean ± s.d. The number of independent biological repeats and technical replicates (*n*) are indicated in the text and figure legends. Unless otherwise stated, the non-parametric Mann–Whitney *U*-test was used to determine statistical significance. All statistical analyses were performed using Prism 9 (GraphPad Software) except for the analysis of RNA sequencing data, where differential gene expression analysis was performed using cuffdiff (v2.2.1) and gene set enrichment analysis with g:Profiler.

### Reporting summary
Further information on research design is available in the Nature Portfolio Reporting Summary linked to this article.

## Data availability
The RNA-sequencing data generated in this study have been deposited in the NCBI Gene Expression Omnibus database under accession code GSE206778. All materials are available upon request from the corresponding authors. Source data are provided with this paper.

## Code availability
The open-source software, tools, and packages used for data analysis in this study, as well as the version of each programme, were trim_galore (v0.4.4), hisat (v2-2.2.1) and cuffdiff (v2.2.1).

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

## Acknowledgements

We thank the Human Research Tissue Bank for processing histological samples. The Human Research Tissue Bank is supported by the NIHR Cambridge Biomedical Research Centre. We would like to thank C. Knox, C. Watson, L. Tredgett, J. Clark, the Ares and Biomed staff for assistance with animal procedures and experimentation. We thank M. Daly, F. Zhang, P. Penttila and the Flow Cytometry Core staff for their technical assistance. We would like to thank C. Perez for assisting with the generation of RNA sequencing libraries. Funding was provided by the Medical Research Council as part of UK Research and Innovation file reference no. MC_UP_1201/18 (G.P.C., R.J.H, N.B., H.W. and A.C.) and Hubrecht Institute (J.I.G.), Nederlandse Organisatie voor Wetenschappelijk Onderzoek (Netherlands Organisation for Scientific Research) project number OCENW.M20.113 (J.S.).

## Author contributions

G.P.C. conceived the study. G.P.C. and R.J.H. designed experiments. R.J.H., N.B. and H.W. performed the experiments. A.C. performed bioinformatic and statistical analyses of RNA sequencing data. J.S. performed data analysis and TUNEL assay. G.P.C., R.J.H., N.B. and J.I.G. wrote the manuscript.

## Competing interests

The authors declare no competing interests.
