## [Peer Review File · Nature Communications]

p53 regulates diverse tissue-specific outcomes to endogenous DNA damage in miceREVIEWER COMMENTS

Reviewer #1 (Remarks to the Author):

In this paper by Hill et al from Gerry Crossan's lab, the authors perform a detailed multi-organ analysis of mice that are deficient in both ERCC1/XPF with and without concomitant deficiency of p53. Their main finding and conclusion is interesting: different tissues respond differently to the same DNA repair defect. The tissue-specific phenotype is dependent on the response to the DNA damage and different tissues behave differently. A lot of this was already known for ERCC1/XPF, but their paper nicely delineates the contribution of the p53 pathway. Overall, the data and conclusions are solid. The writing is good, albeit a bit lengthy.

I have some comments for improvement:

- 1) XPF/ERCC1 is known to be involved in the Fanconi Anemia DNA damage network. Others have previously published crosses between FA mice and p53. There are some clear parallels to the work herein and these papers should be discussed, cited. For example, the rescue of hematopoietic stem cells by p53 modulation in FA is well described: "Cell Stem Cell 2012 Jul 6;11(1):36-49. Bone marrow failure in Fanconi anemia is triggered by an exacerbated p53/p21 DNA damage."
- 2) Introduction and discussion should both be shortened.
- 3) The phenotypic description of the renal phenotype is very superficial. "despite the increase in p21/Cdkn1a and Btg2 expression, there was an increased frequency of proliferating cells". The kidney has many different cell types and it should be pretty straightforward to delineate which cell type was proliferating. My guess is that it must be the proximal tubular epithelium, but this is not shown in the paper.
- 4) In regards to the liver, not enough attention is given to the 2 different modes of polyploidization of hepatocytes. The "normal" physiological mechanism involves failed cytokinesis. Here the authors seem to be describing endoreduplication and massively polyploid individual hepatocyte nodules. The phenomenon described here has nothing to do with physiological, age-related polyploidization.
- 5) The authors state: "It is plausible that polyploidization causes apoptosis". Why is this plausible? There are no reports in the literature supporting this hypothesis.
- 6) What do these mice actually die from? The renal and hepatic dysfunction is quite modest. Any speculation?

Reviewer #2 (Remarks to the Author):

In their manuscript „ p53 regulates diverse tissue-specific outcomes to endogenous DNA damage in mice”, Hill et al. investigate the role of p53 in Ercc1 deficient mice. Ercc1 knockout leads to growth defects and premature aging and has been established as model for XFE progeria. Ercc1 knockout mice provide a mouse model that recapitulates the aging process in an accelerated fashion and due to the versatile functions of the Ercc1-XPF endonuclease in DNA repair processes triggers pathologies in most tissues. The p53 gene is the most frequently mutated tumor suppressor and limited studies have thus far addressed its role in the aging process. Therefore, this manuscript offers new insights that are of general interest.

The authors initially characterize the Ercc1 knockout animals in a mixed 129S6/Sv-C57BL/6 background and determine a normal Mendelian birth rate, reduced lifespan, various tissue pathologies particularly in liver, kidney and bone marrow. These pathologies recapitulate the previously determined pathologies that were shown since the initial establishment of the Ercc1 knockout animals. As a dysfunction of the ERCC1-XPF endonuclease results in unrepaired DNA damage, the authors test whether ablation of Trp53, a central component of the DNA damage response, could mitigate the consequence of a chronic DNA damage response. However, the authors find that p53 actually aggravates a range of Ercc1 ko phenotypes. Mendelian birth rate and body weight were reduced, while

premature death, neuropathology and infertility were unaffected. The hematopoietic defects of *Ercc1* ko were, however, rescued by *Trp53* ablation indicating that the senescence and apoptosis in response to the DNA repair defect in hematopoietic stem cells causes stem cell exhaustion in this tissue. In contrast, liver pathology, particularly with regards to polyploidization was aggravated. These observations indicate that the p53 response to DNA damage could have differential consequences on the aging process in distinct tissues.

The findings are interesting and shed new light on the distinct outcomes of the p53 response in tissue homeostasis. The role of p53 in limiting liver damage by controlling polyploidization are interesting and provide important new insights into liver pathologies associated with cisplatin-mediated chemotherapy. The work is therefore of general interest to the wider community of aging and cancer alike given the central role of *Ercc1* in the former and p53 in the latter.

The experiments appear well conducted and the manuscript is well written. I, therefore, have no requests regarding additional experiments or textual changes.

Reviewer #3 (Remarks to the Author):

In this manuscript, Hill et al. show that p53 plays a tissue-specific role in regulating organ dysfunction in *Ercc1*^{-/-} mice. Specifically, they show that p53 is a regulator of apoptosis in the bone marrow in *Ercc1*^{-/-} mice, whilst it does not affect apoptosis in the kidney and cerebral cortex. In the liver, they show that p53 is responsible for limiting liver polyploidization. Moreover, they suggest a mechanism whereby in the absence of p53, p16 expression is increased, acting as a failsafe mechanism to limit further proliferation and polyploidization of hepatocytes in *Ercc1*^{-/-} mice. The manuscript is very well written and easy to follow. Furthermore, the study is well conducted and the questions addressed by the authors are of high scientific interest. However, I have a few concerns that should be addressed prior to publication, and these are listed below:

General comments:

- Apoptosis should be corroborated using a second marker such as TUNEL assay.
- There is a general lack of consistency in the markers analyzed for each tissue. Whilst the others do not see changes in pathophysiology in the brain and PGCs, what happens to apoptosis and senescence markers in these organs upon p53 deletion?

In the introduction, the authors provide a very thorough background on the role of p53 and how it can vary amongst different tissue types. Although this is very well written and informative, perhaps the introduction could be shortened a little bit, especially the third paragraph.

Supplementary Figure 1:

- The authors refer to serum levels in ALT and proteinuria in the text but the graph is not shown in the figure

Figure 1:

A – Why was a threshold of >5 foci used? Do the authors also see an increase in mean number of gH2AX foci per cell?

The amount of cells containing gH2AX foci seem quite low from the representative images shown considering this is a mouse model with impaired DNA damage repair mechanisms. How do the levels of gH2AX observed here compare to other studies using *Ercc1*^{-/-} mice?

Why are only 3 *Ercc1*^{-/-} mice quantified for bone marrow in A and B but 4 mice in C? The number of control animals also varies for bone marrow analysis. Is there a specific reason for this?

C – Since this is such an important point for this paper (i.e. the fate of different cells upon DNA damage and p53 activation), I suggest that the authors should confirm cell death using a second method, for example, by performing a TUNEL assay, to corroborate these data.

Supplementary Figure 2:

In the kidney of *Ercc1*^{-/-} mice, markers for both apoptosis (Bax and cleaved-caspase 3) and senescence (p21) increase. Could this indicate that different cell types in the kidney are affected differently by absence of *Ercc1*^{-/-} (and thus activation of p53)? Could the authors perform triple immunofluorescence staining for CC3, p21 and different cell type markers in the kidney to assess which cell types are more prone to express pro-apoptotic vs senescent markers? A recent study has

also shown that senescent cells undergo a process known as minority MOMP (miMOMP), whereby some mitochondria in senescent cells become permeabilized (which involves Bax activation) without inducing cell death (see Chapman et al., Research Square, 2022). Could this also happen upon *Ercc1* deletion in kidney cells and possibly explain the upregulation of both p21 and Bax?

Figure 2:

A – E – these figures could probably be put in a supplementary figure. I don't think they fit here very well.

F – Again, I suggest that the authors should corroborate these findings with a TUNEL assay.

F – The authors showed in Supplementary figure 1f that *Ercc1*^{-/-} mice have decreased frequency of hematopoietic stem and progenitor cells. Given that knocking out p53 in *Ercc1*^{-/-} rescues markers of apoptosis, does it also rescue the number of HSPCs?

Supplementary Figure 3:

- The authors speculate the increase in Ki67 despite the increase in p21 expression in *Ercc1*^{-/-} mice kidney is due to regeneration. Whilst this could be the case, it is also possible that, depending on the stage of the cell cycle, some cells are positive for both p21 and Ki67. Ki67 is normally expressed during the G2 phase through to mitosis. I suggest that the authors should assess proliferation using another marker, such as PCNA (expressed during late G1 to S phase). Moreover, the authors should also perform double immunofluorescence staining for p21 and Ki67/PCNA and quantify the frequency of p21+ Ki67/PCNA- cells.

In the text, under results section 3, the authors conclude "Together, these data show that loss of p53 does not drive apoptosis or tissue dysfunction..." – do they mean to say the "p53 does not drive apoptosis..." (not the "loss")?

Figure 3:

- If not p53, what other downstream effectors could be responsible for the upregulation of apoptotic markers and cell-cycle arrest in the kidney of *Ercc1*^{-/-} mice? I.e. what is the mechanism driving these cellular and pathological changes occur in the kidney of *Ercc1*^{-/-} mice?

- Are senescence markers decreased in the brain by p53 deletion in *Ercc1*^{-/-} mice?

- In the text, where authors say "It is therefore unlikely that the activation of p53 plays a role in the observed neurodegenerative changes." This cannot be concluded unless the authors have any behavioral data for these *Ercc1*^{-/-} *Trp53*^{-/-} mice. I suggest this should be removed.

- I also suggest that inflammatory markers be assessed in the brains of these mice. Even though they did not see a decrease in MAC2+ and GFAP+ cells, could the microglia and astrocytes themselves be less pro-inflammatory? If scRNA-seq cannot be performed, the authors could perform methods such as RNA-in situ hybridization for pro-inflammatory markers together with MCA2 or GFAP. Whole-brain qPCR for pro-inflammatory marker might also be informative, however, if the effects are limited to a specific sub-population of cells then differences might not be detected.

Figure 5:

- The authors should also show the protein levels of pro- and anti-apoptotic markers in HSCs to strengthen the data on the apoptotic response.

- Other apoptotic markers should also be shown. For example, can the authors sort these cells and analyze Annexin V by FACS?

The authors suggest that the increase in polyploidization in the liver of *p53*^{-/-} mice is not due to a failure in eliminating these cells as they did not detect an increase in apoptotic hepatocytes nor transcriptional activation of apoptotic factors in *Ercc1*^{-/-} mice. However, is it possible that the authors cannot capture the exact time point in which polyploid hepatocytes undergo apoptosis? I.e. no change in CC3+ hepatocytes means that the majority of hepatocytes already underwent apoptosis at the time the animals were sacrificed? Is it possible to FACS-sort hepatocytes according to DNA content to check if hepatocytes with a higher degree of polyploidy express higher levels of apoptotic markers or are more prone to cell death ?

Figure 6:

- D) The authors should also show mean number of gH2AX foci/ cell.

- G) Is there any significance between WT and *Ercc1*^{-/-}?

- In the text, the authors say – "Therefore, it is likely that loss of the p53-p21 axis allows further hepatocyte proliferation enabling polyploidization." – How does proliferation alone lead to

polyploidization? Is it proliferation without cell division? Can this be demonstrated in vivo?

The authors suggest that "a failsafe p16-dependent cell-cycle arrest mechanism exists to restrain further proliferation and hepatocyte polyploidization in the absence of the p53-p21 axis." – This cannot be directly concluded unless the authors show that knocking out p16 induces further polyploidization.

Figure 7:

- I don't think these data add too much to the manuscript. What was the purpose of showing this and how does it relate to this story?

- The authors say "We hypothesized that inter-strand crosslinks could likely contribute to liver polyploidization..." – in what context? In *Ercc1*^{-/-} mice?

- A-B) the authors say in the text that the responses seen here are similar in magnitude to what was observed spontaneously in *Ercc1*^{-/-} mice. However, they look higher in *Ercc1*^{-/-} mice. I suggest that this should be removed.

REVIEWER COMMENTS

Reviewer #1 (Remarks to the Author):

In this paper by Hill et al from Gerry Crossan's lab, the authors perform a detailed multi-organ analysis of mice that are deficient in both ERCC1/XPF with and without concomitant deficiency of p53. Their main finding and conclusion is interesting: different tissues respond differently to the same DNA repair defect. The tissue-specific phenotype is dependent on the response to the DNA damage and different tissues behave differently. A lot of this was already known for ERCC1/XPF, but their paper nicely delineates the contribution of the p53 pathway. Overall, the data and conclusions are solid. The writing is good, albeit a bit lengthy.

Our response: we are pleased the Reviewer finds this work interesting and we thank them for their comments on the robustness of the data and conclusions. We have taken the advice on board and made the text more succinct.

I have some comments for improvement:

1) XPF/ERCC1 is known to be involved in the Fanconi Anemia DNA damage network. Others have previously published crosses between FA mice and p53. There are some clear parallels to the work herein and these papers should be discussed, cited. For example, the rescue of hematopoietic stem cells by p53 modulation in FA is well described: "Cell Stem Cell 2012 Jul 6;11(1):36-49. Bone marrow failure in Fanconi anemia is triggered by an exacerbated p53/p21 DNA damage."

Our response: We thank the reviewer for pointing out this omission. Indeed, our current results align very well with those of Ceccaldi et al, Cell Stem Cell, 2012 who show a rescue of the *Fancd2*^{-/-} HSC defect in *Fancd2*^{-/-} *p53*^{-/-} mice, and also our own work where we show that the severe 600-fold HSC defect and pancytopenia of *Aldh2*^{-/-} *Fancd2*^{-/-} mice are completely rescued in *Aldh2*^{-/-} *Fancd2*^{-/-} *p53*^{-/-} mice (Garaycochea et al, Nature, 2018)¹⁻⁴. In addition, Zhang et al, Stem Cell Res, 2013, observed that the FA HSC defect is not rescued in *Fancd2*^{-/-} *p21*^{-/-} mice and conclude 'that other p53-targeted factors, not p21, mediate the progressive elimination of HSPC in FA', which is in agreement with the apoptotic response that we observe in *Ercc1*^{-/-} HSCs. We have now added this to the Discussion (lines 453-457).

2) Introduction and discussion should both be shortened.

Our response: as suggested by the Reviewer, we have shortened the Introduction by 20% to 820 words, and the Discussion by 29% to 1108 words.

3) The phenotypic description of the renal phenotype is very superficial. "despite the increase in p21/Cdkn1a and Btg2 expression, there was an increased frequency of proliferating cells". The kidney has many different cell types and it should be pretty straightforward to delineate which cell type was proliferating. My guess it that it must be the proximal tubular epithelium, but this is not shown in the paper.

Our response: we agree that the kidney is a very complex organ, in terms of function, architecture and cellular complexity. Previous work shows that p53 activation mediates cisplatin-induced nephrotoxicity. Interestingly, the p53 checkpoint response is also complex, cell-type specific and differs along the nephron: proximal tubule-specific p53 KO are resistant to acute kidney injury, whereas p53 loss in other tubular segments (distal tubules, loops of Henle and medullary collecting ducts) increased sensitivity to acute kidney injury⁵. To better delineate checkpoint responses across the nephron in *Ercc1*^{-/-} and *Ercc1*^{-/-} *p53*^{-/-} mice, we attempted to perform immunostaining of AQP1 (proximal tubule, descending limb), Calb1 (distal tubule) or AQP2 (connecting tubule and collecting duct) together with Ki-67, PCNA or p21, however our attempts were unsuccessful. In our hands the PCNA and p21 antibodies gave a much cleaner signal by IHC than immunofluorescence, but this precludes double and triple staining. Both the PCNA and p21 immunohistochemistry showed a clear induction in *Ercc1*^{-/-} kidneys; the p21 staining was reduced by the loss of p53, although this difference was not significant (**Supplementary Figure 3e**). We observed that signal for PCNA, CC3 and p21 occurred in the cortex, in cells that show histological features consistent with the proximal tubule - which is not surprising. However, cells within the medulla also stained positive which may represent cells of the thin descending limb (**Rebuttal Figure 1**). Therefore, cells of the proximal tubule are proliferating in *Ercc1*^{-/-} and *Ercc1*^{-/-} *p53*^{-/-} kidneys but cell proliferation is not restricted to this cell type. In this regard, a recent preprint shows a requirement for *Ercc1* in podocytes: podocyte-specific deletion of *Ercc1* leads to proteinuria, podocyte loss, glomerulosclerosis, renal insufficiency, and reduced lifespan⁶. Thoroughly

dissecting the function of *Ercc1* in this complex organ would require cell-type specific KOs (e.g. analogous to the podocyte *Ercc1* KO), together with scRNA-seq to map transcriptional changes in specific cell types. However, we think that our current work sets the foundation for future studies to fully understand the basis of the checkpoint response and renal pathobiology in *Ercc1* mice.

Rebuttal Figure 1 - Immunohistochemistry staining of the kidney of *Ercc1*^{-/-} mice for p21 (top) or PCNA (bottom) showing staining within the renal medulla.

4) In regards to the liver, not enough attention is given to the 2 different modes of polyploidization of hepatocytes. The “normal” physiological mechanism involves failed cytokinesis. Here the authors seem to be describing endoreduplication and massively polyploid individual hepatocyte nodules. The phenomenon described here has nothing to do with physiological, age-related polyploidization.

Our response: we agree with the reviewer that in the human liver, indeed many polyploid hepatocytes are binucleated arising from failed cytokinesis. It is likely that failed cytokinesis is most associated with ageing whilst endoreduplication has a stronger association with damage and regeneration - we have added this caveat to the results section (line 353, 357-9)⁷. However, in aged human liver, mononucleated polyploid hepatocytes which may arise from endoreplication are also common. For example, in 80-year old livers 20% of hepatocytes are binucleated 2x2n, 20% are mononucleated 1x4n and 3% are 1x8n (see Heinke et al., 2022, Cell Systems 13, 499–507)⁸. This argues that both mechanisms of polyploidization are active and physiological in aged liver. The FACS analysis of ploidy in Figure 6a is done on nuclei (not cells) so indeed the massively polyploid events (16n, 32n, 64n) are individual nuclei, in agreement with the microscopy pictures. Indeed, we think these massive nuclei are arising by endoreplication, which has been shown to be a response to DNA damage in certain settings, for example in p53-null MEFs⁹. In *Ercc1*^{-/-} p53^{-/-} livers, we believe this is a consequence of the chronic exposure to endogenous DNA damage (Figure 6c) combined with an abnormal cell-cycle checkpoint (Figure 6e). We have now made a clearer distinction between the 2 modes of polyploidization around these results (lines 357-59).

5) The authors state: “It is plausible that polyploidization causes apoptosis”. Why is this plausible? There are no reports in the literature supporting this hypothesis.

Our response: we thank the Reviewer for pointing this out, we have removed this statement and replaced it with the following text in line 326: “A canonical function of p53 is the activation of apoptosis, however, we did not detect an increased frequency of apoptotic hepatocytes (CC3+) nor transcriptional activation of apoptotic factors in *Ercc1*^{-/-} (Figure 1c, 2f and Supplementary Figure 2a).”

6) What do these mice actually die from? The renal and hepatic dysfunction is quite modest. Any speculation?

Our response: Liver failure is considered the primary cause of death in *Ercc1*^{-/-} single KO mice^{10,11}. The most compelling evidence comes from an experiment in which *Ercc1*^{-/-} mice were crossed with mice carrying an ERCC1 transgene expressed under the control of the liver-specific, transthyretin (TTR) promoter¹¹. This alleviated runting, reversed liver polyploidy, restored liver function and dramatically extended the lifespan of *Ercc1*^{-/-} mice. However, *Ercc1*^{-/-} +TTR-ERCC1 transgenic mice displayed evidence of renal dysfunction (significantly elevated serum

creatinine and proteinuria) and abnormal renal histology (glomerulosclerosis, hyaline casts, renal tubular epithelial anisokaryosis, karyomegaly, hyperchromasia, pyknosis and karyorrhexis). Ultimately these mice succumbed to renal failure by 12-13 weeks¹¹. Given that the *Ercc1*^{-/-} *p53*^{-/-} mice we describe here have indistinguishable survival compared to *Ercc1*^{-/-} controls and similar liver and kidney dysfunction, we speculate that double mutants also succumb to liver failure as the primary cause of death.

Reviewer #2 (Remarks to the Author):

In their manuscript “p53 regulates diverse tissue-specific outcomes to endogenous DNA damage in mice”, Hill et al. investigate the role of p53 in Ercc1 deficient mice. Ercc1 knockout leads to growth defects and premature aging and has been established as model for XFE progeria. Ercc1 knockout mice provide a mouse model that recapitulates the aging process in an accelerated fashion and due to the versatile functions of the Ercc1-XPF endonuclease in DNA repair processes triggers pathologies in most tissues. The p53 gene is the most frequently mutated tumor suppressor and limited studies have thus far addressed its role in the aging process. Therefore, this manuscript offers new insights that are of general interest.

The authors initially characterize the Ercc1 knockout animals in a mixed 129S6/Sv-C57BL/6 background and determine a normal Mendelian birth rate, reduced lifespan, various tissue pathologies particularly in liver, kidney and bone marrow. These pathologies recapitulate the previously determined pathologies that were shown since the initial establishment of the Ercc1 knockout animals. As a dysfunction of the ERCC1-XPF endonuclease results in unrepaired DNA damage, the authors test whether ablation of Trp53, a central component of the DNA damage response, could mitigate the consequence of a chronic DNA damage response. However, the authors find that p53 actually aggravates a range of Ercc1 ko phenotypes. Mendelian birth rate and body weight were reduced, while premature death, neuropathology and infertility were unaffected. The hematopoietic defects of Ercc1 ko were, however, rescued by Trp53 ablation indicating that the senescence and apoptosis in response to the DNA repair defect in hematopoietic stem cells causes stem cell exhaustion in this tissue. In contrast, liver pathology, particularly with regards to polyploidization was aggravated. These observations indicate that the p53 response to DNA damage could have differential consequences on the aging process in distinct tissues.

The findings are interesting and shed new light on the distinct outcomes of the p53 response in tissue homeostasis. The role of p53 in limiting liver damage by controlling polyploidization are interesting and provide important new insights into liver pathologies associated with cisplatin-mediated chemotherapy. The work is therefore of general interest to the wider community of aging and cancer alike given the central role of Ercc1 in the former and p53 in the latter.

The experiments appear well conducted and the manuscript is well written. I, therefore, have no requests regarding additional experiments or textual changes.

Our response: We thank the reviewer for their feedback. We are glad that the reviewer finds the experiments well conducted, the manuscript well written and of interest to the wider scientific community of aging and cancer.

Reviewer #3 (Remarks to the Author):

In this manuscript, Hill et al. show that p53 plays a tissue-specific role in regulating organ dysfunction in *Ercc1*^{-/-} mice. Specifically, they show that p53 is a regulator of apoptosis in the bone marrow in *Ercc1*^{-/-} mice, whilst it does not affect apoptosis in the kidney and cerebral cortex. In the liver, they show that p53 is responsible for limiting liver polyploidization. Moreover, they suggest a mechanism whereby in the absence of p53, p16 expression is increased, acting as a failsafe mechanism to limit further proliferation and polyploidization of hepatocytes in *Ercc1*^{-/-} mice. The manuscript is very well written and easy to follow. Furthermore, the study is well conducted and the questions addressed by the authors are of high scientific interest. However, I have a few concerns that should be addressed prior to publication, and these are listed below:

General comments:

- Apoptosis should be corroborated using a second marker such as TUNEL assay.

Our response: We agree with the reviewer that this would add complementary data to that presented in the manuscript and we have now performed the TUNEL assay and the data is shown in **Supplementary Figure 2b**.

- There is a general lack of consistency in the markers analysed for each tissue. Whilst the others do not see changes in pathophysiology in the brain and PGCs, what happens to apoptosis and senescence markers in these organs upon p53 deletion?

Our response: Firstly, we have performed p21 staining (a senescence marker) in the brain and find that loss of p53 does not significantly affect the induction of senescence (**Supplementary Figure 4c**). However, the brain of *Ercc1*-deficient mice has been extensively studied previously and requires specialised models to allow firm conclusions to be drawn. The liver and kidney pathologies described in *Ercc1*^{-/-} mice (which are altered by the loss of p53) are likely to play a major role in the brain. Indeed, in patients with liver and kidney dysfunction hepatic and uraemic encephalopathy is a major cause of morbidity. Therefore, to assess the consequence of p53 loss on the brain one would want to employ a brain specific deletion (as has been used previously). The data presented in the current manuscript show that loss of p53 does not affect the pathophysiology but we would not want to draw conclusions beyond this.

We have not measured apoptosis and senescence in the PGC pool of *Ercc1*^{-/-} or *Ercc1*^{-/-} *p53*^{-/-} embryos due to the technical complexity of this experiment. Our previous work has shown that an increased frequency of *Ercc1*^{-/-} PGCs undergo apoptosis (Hill & Crossan 2019) - this was a major undertaking as mice must be set up for timed matings, culled, embryos harvested, embryonic gonads dissected and stained. This becomes much more complex when we consider that *Ercc1*^{-/-} *p53*^{-/-} embryos are born at a sub-Mendelian ratio (3%, **Figure 2a**). The average litter size is 6 pups, therefore if we were to generate 8 *Ercc1*^{-/-} *p53*^{-/-} embryos (4 each for p21 and CC3) we would need a total of 266 embryos, which equates to 44 pregnant females. This is not feasible given the constraints and expense of *in vivo* studies.

In the introduction, the authors provide a very thorough background on the role of p53 and how it can vary amongst different tissue types. Although this is very well written and informative, perhaps the introduction could be shortened a little bit, especially the third paragraph.

Our response: Whilst we appreciate that the reviewer thinks the introduction is very thorough, we agree and have now shortened the introduction by 20% to 820 words.

Supplementary Figure 1:

- The authors refer to serum levels in ALT and proteinuria in the text but the graph is not shown in the figure

Our response: We thank the reviewer for highlighting this error. We have now included serum ALT and albumin at **Supplementary Figure 1d**. To provide evidence of proteinuria we have taken urine samples from 3 wildtype and 3 *Ercc1*^{-/-} mice and run it on a SDS-PAGE before staining it with Coomassie (**Supplementary Figure 1f**). This clearly shows that *Ercc1*-deficient mice have large amounts of albumin in the urine compared to wildtype littermates.

Figure 1:

A – Why was a threshold of >5 foci used? Do the authors also see an increase in mean number of γ H2A.X foci per cell?

Our response: We thank the reviewer for their comment concerning the use of a threshold of γ H2A.X. Thresholds are frequently used for γ H2A.X quantification as presenting the mean alone risks masking the induction of DNA damage with the “background” frequency of γ H2A.X (e.g. DSBs generated by topoisomerase etc.). We have now quantified the mean number of γ H2A.X per cell and find that in addition to having a greater proportion of cells with >5 foci in liver, kidney and bone marrow (Figure 1a), *Ercc1*^{-/-} mice also have a cells with a higher mean number of γ H2A.X foci when compared to wildtype (**Supplementary Figure 2a**).

The amount of cells containing γ H2A.X foci seem quite low from the representative images shown considering this is a mouse model with impaired DNA damage repair mechanisms. How do the levels of γ H2A.X observed here compare to other studies using *Ercc1*^{-/-} mice?

Our response: We thank the reviewer for this comment. The frequency of cells that stain positive for γ H2A.X is indeed low if it is compared to tissue culture cells. It is worth noting that in these *in vivo* sample we are measuring the burden of endogenous DNA damage and that the animals have intact DNA damage checkpoints which is not the case for most tissue culture cell lines. It is likely that *in vivo* cells with DNA damage are rapidly eliminated or the damage is repaired; however, it is unlikely that they persist or go through additional rounds of replication which is more likely to occur in cancer tissue culture cell lines. There have been remarkably few studies quantifying the frequency of γ H2A.X in the absence of *Ercc1*. In Vermeij et al., approximately 2 γ H2A.X positive purkinje neurons were detected per section although this mouse carried one null and one hypomorphic allele of *Ercc1*¹². Therefore, our finds are in keeping with a singular previous report of γ H2A.X in *Ercc1*^{-/-}.

Why are only 3 *Ercc1*^{-/-} mice quantified for bone marrow in A and B but 4 mice in C? The number of control animals also varies for bone marrow analysis. Is there a specific reason for this?

Our response: We thank the reviewer for this comment. The difference in the number of animals used is entirely technical. We ran bone marrow staining for CC3 each time we stained either the liver or kidney for CC3. This was as a positive control to ensure that the staining worked and that we could detect CC3+ cells in the bone marrow. This leads to the discrepancy in numbers between Figure 1 A, B and C.

C – Since this is such an important point for this paper (i.e. the fate of different cells upon DNA damage and p53 activation), I suggest that the authors should confirm cell death using a second method, for example, by performing a TUNEL assay, to corroborate these data.

Our response: We have now performed the TUNEL assay and the data is shown in **Supplementary Figure 2b**. In agreement with the CC3 staining both the bone marrow and the kidney for *Ercc1*^{-/-} mice have an induction in the frequency of apoptotic cells.

Supplementary Figure 2:

In the kidney of *Ercc1*^{-/-} mice, markers for both apoptosis (Bax and cleaved-caspase 3) and senescence (p21) increase. Could this indicate that different cell types in the kidney are affected differently by absence of *Ercc1*^{-/-} (and thus activation of p53)? Could the authors perform triple immunofluorescence staining for CC3, p21 and different cell type markers in the kidney to assess which cell types are more prone to express pro-apoptotic vs senescent markers? A recent study has also shown that senescent cells undergo a process known as minority MOMP (miMOMP), whereby some mitochondria in senescent cells become permeabilized (which involves Bax activation) without inducing cell death (see Chapman et al., Research Square, 2022). Could this also happen upon *Ercc1* deletion in kidney cells and possibly explain the upregulation of both p21 and Bax?

Our response: We agree that it would be informative to know if the same cells stain positive for CC3 and p21. We did attempt to perform triple immunostaining however encountered technical issues which made our attempts unsuccessful. In our hands the p21 antibody gave a much cleaner signal by IHC than immunofluorescence, but this precludes double and triple staining. We did stain the kidney for p21 (**Supplementary Figure 3e**) and found a clear induction in *Ercc1*^{-/-} that was reduced by the loss of p53, but not significantly. The signal for both PCNA, CC3 and p21 occurred in cells that show histological features consistent with the proximal tubule - which is not surprising. However, cells within the medulla also stained positive which may represent cells of the thin descending limb (See

Reviewer 1 point 3). We think that this work will be the foundation of future studies to fully understand the basis of the renal pathobiology. We agree with the reviewer's insightful suggestion that miMOMP may be contributing to our phenotype and now discuss this possibility in the text (line 484) "Recent work has shown that senescent cells may undergo mitochondrial outer membrane permeabilization which may contribute to the observed activation of senescence and apoptosis markers in *Ercc1*^{-/-} kidneys¹³".

Figure 2:

A – E – these figures could probably be put in a supplementary figure. I don't think they fit here very well.

Our response: We are open to moving this data to supplementary however, neither of the other reviewers have raised this concern. Therefore, in the resubmission we have kept the data in place but are willing to move this if it is considered necessary.

F – Again, I suggest that the authors should corroborate these findings with a TUNEL assay.

Our response: We agree with the reviewer that validating that the CC3 staining is indeed measuring apoptosis is important. However, the cleaved caspase-3 antibody is widely used and accepted as a *bone fide* marker of apoptosis. We have gone further than this and performed the TUNEL assay on the bone marrow, liver and kidney of wildtype and *Ercc1*-deficient mice. These data confirm an induction in apoptosis in both the kidney and bone marrow of *Ercc1*-deficient mice but not in the liver (Supplementary Figure 2b) - these findings are consistent with the CC3 staining. Furthermore, we show that the kidney of *Ercc1*^{-/-} and *Ercc1*^{-/-}*p53*^{-/-} mice have transcriptional changes consistent with the CC3 staining (Figure Figure 3a and Supplementary Figure 3b). Furthermore, no increase in apoptosis was detected in the liver by CC3 staining, TUNEL assay or transcriptionally (Figure 1c, Supplementary Figure 2c and Rebuttal Figure 2). Together, we believe that this comprehensively addresses the concern that CC3 is not measuring apoptosis.

F – The authors showed in Supplementary figure 1f that *Ercc1*^{-/-} mice have decreased frequency of hematopoietic stem and progenitor cells. Given that knocking out p53 in *Ercc1*^{-/-} rescues markers of apoptosis, does it also rescue the number of HSPCs?

Our response: We agree with the reviewer that the suppression of apoptosis in the bone marrow of *Ercc1*^{-/-} *p53*^{-/-} mice may rescue the HSPC number. We show in Figure 5 b-d that indeed the frequency of HSPCs (lineage-c-Kit+Sca-1+) and HSCs (lineage-c-Kit+Sca-1+CD41-CD48-CD150+) are rescued in *Ercc1*^{-/-} *p53*^{-/-} mice when compared to *Ercc1*^{-/-}.

Supplementary Figure 3:

- The authors speculate the increase in Ki67 despite the increase in p21 expression in *Ercc1*^{-/-} mice kidney is due to regeneration. Whilst this could be the case, it is also possible that, depending on the stage of the cell cycle, some cells are positive for both p21 and Ki67. Ki67 is normally expressed during the G2 phase through to mitosis. I suggest that the authors should assess proliferation using another marker, such as PCNA (expressed during late G1 to S phase). Moreover, the authors should also perform double immunofluorescence staining for p21 and Ki67/PCNA and quantify the frequency of p21+ Ki67/PCNA- cells.

Our response: We thank the reviewer for raising this important point. We have now performed staining for PCNA and p21 in the kidney (**Supplementary Figure 3d and e**). We find that there is a significant induction in the frequency of PCNA+ cells in *Ercc1*^{-/-} mice and this is not suppressed by the loss of p53. This is in agreement with the proliferation data obtained by staining for Ki67 (**Supplementary Figure 3c**). This data suggests that the loss of *Ercc1* does lead to increased proliferation in the kidney and that loss of p53 does not subvert this process. We also find that there is an increase in the frequency of p21+ cells in *Ercc1*^{-/-}, which again is not suppressed by the loss of p53. This is in agreement with the RT-qPCR expression data (**Figure 3a**) and previous reports of p53-independent activation of p21¹⁴. As discussed earlier, we only obtained reliable p21 staining by traditional IHC and whilst this is very robust it precludes double staining with Ki67 and PCNA.

In the text, under results section 3, the authors conclude “Together, these data show that loss of p53 does not drive apoptosis or tissue dysfunction...” – do they mean to say the “p53 does not drive apoptosis...” (not the “loss”)?

Our response: We thank the reviewer for highlighting this error that we have now corrected (line 224).

Figure 3:

- If not p53, what other downstream effectors could be responsible for the upregulation of apoptotic markers and cell-cycle arrest in the kidney of *Ercc1*^{-/-} mice? I.e. what is the mechanism driving these cellular and pathological changes occur in the kidney of *Ercc1*^{-/-} mice?

Our response: This is a very interesting question that will form the basis of future studies. From our study it seems that p21 plays a major role in the pathobiology in the kidney of *Ercc1*^{-/-} mice. Whilst the activation of p21 in response to DNA damage by p53 is well understood, it has long been appreciated that there are multiple p53-independent routes for the activation of p21 (reviewed in Gartel *et al.*¹⁵). It is worth noting that we do observe non-significant reduction in p21 mRNA and protein (**Figure 3a** and **Supplementary Figure 3e**) suggesting that some component of p21 activation may be dependent upon p53. Going forward we will need to establish which of these is responsible for the activation of p21 in the kidney of *Ercc1*-deficient mice.

There is substantial evidence for the activation of apoptosis independently of p53^{16,17}. Furthermore, in certain DNA repair deficiencies (e.g. Seckel syndrome) the loss of p53 exacerbates apoptosis. p53-independent apoptosis may be a result of cells carrying DNA damage undergoing further rounds of replication which can result in even more DNA damage or mitotic catastrophe. It is worth noting that we see elevated levels of PCNA+ and Ki-67+ cells in the kidney of *Ercc1*^{-/-} and *Ercc1*^{-/-}*p53*^{-/-} mice.

We have now added the following text to the discussion “Even in the absence of p53 we detect a significant increase the frequency of p21+ cells. p21 activation is often downstream of p53 however, p53-independent activation of p21 is well documented. Determining the molecular basis of p21 activation in *Ercc1*^{-/-} *Trp53*^{-/-} kidneys will be an important area of future study. We also found activation of apoptotic factors and an increase in the proportion of apoptotic cells that persisted even when p53 was lost. This suggests that p53-independent apoptosis makes an important contribution to tissue homeostasis in the kidney. p53-independent apoptosis in response to DNA damage has previously been described and it will be interesting to elucidate this mechanism in future work.”

- Are senescence markers decreased in the brain by p53 deletion in *Ercc1*^{-/-} mice?

Our response: We thank the reviewer for this comment and we have now stained the brain of *Ercc1*^{-/-} and *Ercc1*^{-/-}*p53*^{-/-} for p21. p21 is a canonical marker of senescence in dividing cells but has also been shown to be expressed in senescent neurons. We found an induction of p21+ cells in both the cerebral cortex and cerebellum of *Ercc1*^{-/-} mice (**Supplementary Figure 4c**). However, loss of p53 did not alter the frequency of p21+ cells suggesting p53-independent activation of p21 which has been reported previously¹⁸.

- In the text, where authors say “It is therefore unlikely that the activation of p53 plays a role in the observed neurodegenerative changes.” This cannot be concluded unless the authors have any behavioral data for these *Ercc1*^{-/-} *Trp53*^{-/-} mice. I suggest this should be removed.

Our response: We thank the reviewer for pointing out this error and we have now removed this sentence.

- I also suggest that inflammatory markers be assessed in the brains of these mice. Even though they did not see a decrease in MAC2+ and GFAP+ cells, could the microglia and astrocytes themselves be less pro-inflammatory? If scRNA-seq cannot be performed, the authors could perform methods such as RNA-in situ hybridization for pro-inflammatory markers together with MCA2 or GFAP. Whole-brain qPCR for pro-inflammatory marker might also be

informative, however, if the effects are limited to a specific sub-population of cells then differences might not be detected.

Our response: We thank the reviewer for raising this point. We agree that despite not seeing a reduction in the frequency of MAC2+ or GFAP+ we should measure pro-inflammatory markers to ascertain if loss of p53 affects their expression. We have now performed RT-qPCR from whole brain samples on a panel of inflammatory markers shown in **Supplementary Figure 4d**. This panel of inflammatory, senescence associated and disease associated microglia (DAM) genes was chosen as they have previously been implicated in the neuropathological changes observed in *Ercc1*-deficient mice^{19,20}. These data reveal that whilst a subset of inflammatory markers are upregulated in *Ercc1*^{-/-} the loss of p53 does not significantly reduce their expression.

Figure 5:

- The authors should also show the protein levels of pro- and anti-apoptotic markers in HSCs to strengthen the data on the apoptotic response.
- Other apoptotic markers should also be shown. For example, can the authors sort these cells and analyze Annexin V by FACS?

Our response: We agree with the reviewer that it would be an excellent experiment to measure pro- or anti apoptotic markers at the protein level or by flow cytometry in HSCs. These experiments are very difficult to conduct given the scarcity of HSCs (both in wildtype but even more so in the *Ercc1*-deficient mice). From the entire bone marrow of a wild type mouse we are only likely to obtain 1000-2000 HSCs following FACS with even fewer in *Ercc1*^{-/-} (100-200 per mouse). This makes downstream applications such as immunoblot extremely challenging. For immunoblot in HSCs, 2000-5000 cells are required therefore, we would require at least 10 *Ercc1*^{-/-}²¹. Given the mendelian ratio of 25% we would need to breed 40 mice for a single western blot making this experiment unfeasible. However, we think that the transcriptional and phenotypic rescue provide compelling evidence for the role of p53 in the loss of HSCs in *Ercc1*^{-/-} mice. We have now better contextualised our data explaining how it fits with previously published work showing the role of p53 in HSC loss (See R1 point 1). This previously published work shows that ablation of p53 rescues the HSC defect of *Fancc2* deficient mice but that p21 ablation does not. Given that *Ercc1* acts together with *Fancc2* in the FA pathway our transcriptional and genetic data strongly argue that p53 plays an analogous role in *Ercc1*-deficient HSCs as other FA pathway components. We have now added this to the discussion.

The authors suggest that the increase in polyploidization in the liver of p53^{-/-} mice is not due to a failure in eliminating these cells as they did not detect an increase in apoptotic hepatocytes nor transcriptional activation of apoptotic factors in *Ercc1*^{-/-} mice. However, is it possible that the authors cannot capture the exact time point in which polyploid hepatocytes undergo apoptosis? I.e. no change in CC3+ hepatocytes means that the majority of hepatocytes already underwent apoptosis at the time the animals were sacrificed? Is it possible to FACS-sort hepatocytes according to DNA content to check if hepatocytes with a higher degree of polyploidy express higher levels of apoptotic markers or are more prone to cell death ?

Our response: We thank the reviewer for raising this important point. We have now added this caveat to the main text (line 329) “However, despite not detecting evidence of apoptosis in hepatocytes (CC3+ or a transcriptional signal) we cannot exclude a role for apoptosis (e.g. at an earlier developmental stage).” We agree that it would be informative to compare polyploid with diploid hepatocytes. However, this is technically challenging. We isolate nuclei in order to assess polyploidisation therefore precluding the downstream analysis of cytoplasmic mRNA.

Figure 6:

- D) The authors should also show mean number of γH2A.X foci/ cell.

Our response: We have now included this data in **Supplementary Figure 7c**.

- G) Is there any significance between WT and *Ercc1*^{-/-}?

Our response: We thank the reviewer for highlighting this glaring omission. The difference is statistically significant (p=0.005) and we have added this to the figure. This finding is consistent with the data shown in **Supplementary Figure 2c**.

- In the text, the authors say – “Therefore, it is likely that loss of the p53-p21 axis allows further hepatocyte proliferation enabling polyploidization.” – How does proliferation alone lead to polyploidization? Is it proliferation without cell division? Can this be demonstrated in vivo?

The authors suggest that “a failsafe p16-dependent cell-cycle arrest mechanism exists to restrain further proliferation and hepatocyte polyploidization in the absence of the p53-p21 axis.” – This cannot be directly concluded unless the authors show that knocking out p16 induces further polyploidization.

Our response: We thank the reviewer for raising this point. We have now expanded the discussion of how polyploidisation may occur and contextualise our results (see also R1 point 4). However, given that we observe an increase in mononucleated hepatocytes with increased chromosome numbers >64n, this is likely the result of endoreduplication which is consistent with our observed increase in Ki67+ and PCNA+ hepatocytes. We have now re-written to make the conclusion more precise and supported by the data “In the absence of the p53-p21 axis, we see engagement of p16 and arrested proliferation. Given the documented role of p16 in cell cycle regulation, this may represent a failsafe mechanism to prevent proliferation and endoreduplication.”

Figure 7:

- I don't think these data add too much to the manuscript. What was the purpose of showing this and how does it relate to this story?

- The authors say “We hypothesized that inter-strand crosslinks could likely contribute to liver polyploidization...” – in what context? In *Ercc1*^{-/-} mice?

- A-B) the authors say in the text that the responses seen here are similar in magnitude to what was observed spontaneously in *Ercc1*^{-/-} mice. However, they look higher in *Ercc1*^{-/-} mice. I suggest that this should be removed.

Our response: We agree with the reviewer and have now added Figure 7 to supplementary information.

References

1. Ceccaldi, R. *et al.* Bone marrow failure in Fanconi anemia is triggered by an exacerbated p53/p21 DNA damage response that impairs hematopoietic stem and progenitor cells. *Cell Stem Cell* **11**, 36-49 (2012).
2. Zhang, Q.S. *et al.* Fancd2 and p21 function independently in maintaining the size of hematopoietic stem and progenitor cell pool in mice. *Stem Cell Res* **11**, 687-92 (2013).
3. Garaycochea, J.I. *et al.* Genotoxic consequences of endogenous aldehydes on mouse haematopoietic stem cell function. *Nature* **489**, 571-5 (2012).
4. Garaycochea, J.I. *et al.* Alcohol and endogenous aldehydes damage chromosomes and mutate stem cells. *Nature* **553**, 171-177 (2018).
5. Zhang, D. *et al.* Tubular p53 regulates multiple genes to mediate AKI. *J Am Soc Nephrol* **25**, 2278-89 (2014).
6. Braun, F. *et al.* Loss of genome maintenance accelerates podocyte damage. *bioRxiv*, 2020.09.13.295303 (2022).
7. Celton-Morizur, S. & Desdouets, C. Polyploidization of liver cells. *Adv Exp Med Biol* **676**, 123-35 (2010).
8. Heinke, P. *et al.* Diploid hepatocytes drive physiological liver renewal in adult humans. *Cell Syst* **13**, 499-507 e12 (2022).
9. Davoli, T., Denchi, E.L. & de Lange, T. Persistent telomere damage induces bypass of mitosis and tetraploidy. *Cell* **141**, 81-93 (2010).
10. McWhir, J., Selfridge, J., Harrison, D.J., Squires, S. & Melton, D.W. Mice with DNA repair gene (ERCC-1) deficiency have elevated levels of p53, liver nuclear abnormalities and die before weaning. *Nat Genet* **5**, 217-24 (1993).
11. Selfridge, J., Hsia, K.T., Redhead, N.J. & Melton, D.W. Correction of liver dysfunction in DNA repair-deficient mice with an ERCC1 transgene. *Nucleic Acids Res* **29**, 4541-50 (2001).
12. Vermeij, W.P. *et al.* Restricted diet delays accelerated ageing and genomic stress in DNA-repair-deficient mice. *Nature* **537**, 427-431 (2016).
13. Victorelli, S. *et al.* Author Correction: Apoptotic stress causes mtDNA release during senescence and drives the SASP. *Nature* (2024).
14. Megyesi, J., Udvarhelyi, N., Safirstein, R.L. & Price, P.M. The p53-independent activation of transcription of p21 WAF1/CIP1/SDI1 after acute renal failure. *Am J Physiol* **271**, F1211-6 (1996).

15. Gartel, A.L. & Tyner, A.L. Transcriptional regulation of the p21((WAF1/CIP1)) gene. *Exp Cell Res* **246**, 280-9 (1999).
16. Strasser, A., Harris, A.W., Jacks, T. & Cory, S. DNA damage can induce apoptosis in proliferating lymphoid cells via p53-independent mechanisms inhibitable by Bcl-2. *Cell* **79**, 329-39 (1994).
17. Park, H.Y. *et al.* Induction of p53-Independent Apoptosis and G1 Cell Cycle Arrest by Fucoidan in HCT116 Human Colorectal Carcinoma Cells. *Mar Drugs* **15**(2017).
18. Macleod, K.F. *et al.* p53-dependent and independent expression of p21 during cell growth, differentiation, and DNA damage. *Genes Dev* **9**, 935-44 (1995).
19. Geirsdottir, L. *et al.* Cross-Species Single-Cell Analysis Reveals Divergence of the Primate Microglia Program. *Cell* **179**, 1609-1622 e16 (2019).
20. Zhang, X. *et al.* Intrinsic DNA damage repair deficiency results in progressive microglia loss and replacement. *Glia* **69**, 729-745 (2021).
21. Cai, X., Zheng, Y. & Speck, N.A. A Western Blotting Protocol for Small Numbers of Hematopoietic Stem Cells. *J Vis Exp* (2018).

REVIEWERS' COMMENTS

Reviewer #1 (Remarks to the Author):

The authors have addressed my main concerns.

I have a couple of remaining comments:

- 1) I don't like the term "segmental". This is not generally used in the context it is employed for here. Segmental sounds like an allusion to the fruit fly body plan. The mouse does not have "segments". "Tissue specific" is the term that should be used.
- 2) The introduction is still too verbose for my taste.
- 3) The GDF-15 hypothesis was a dead end and should go into the supplementary data.

Reviewer #3 (Remarks to the Author):

The authors have addressed the majority of my suggestions. I recommend that this article should now be published.

Point by point response to reviewers

Reviewer #1 (Remarks to the Author):

The authors have addressed my main concerns.

Our response: We are glad that our revisions have satisfied the reviewer

I have a couple of remaining comments:

1) I don't like the term "segmental". This is not generally used in the context it is employed for here. Segmental sounds like an allusion to the fruit fly body plan. The mouse does not have "segments". "Tissue specific" is the term that should be used.

Our response: The use of the word "segmental" to describe Ercc1-deficient mice is common, has occurred for more than 10 years and predates our entry into the field. E.g. The paper entitled "Broad segmental progeroid changes in short-lived Ercc1- Δ 7 mice" clearly describes the phenotype as segmental¹. Other recent examples include Vougioukalaki et al. in which the Ercc1-deficient mouse is used to probe progeria, "explaining heterogeneity in organ ageing and the segmental nature of DNA-repair-deficient progerias."² In Niederhofer et al. the text states "The segmental nature of these progeroid syndromes is consistent..."³. The title of our paper describes "tissue-specific" as the reviewer suggests, however, we use do use segmental especially referring to previously described aspects of the Ercc1-deficient phenotype – the same wording that was used by the authors of those reports.

2) The introduction is still too verbose for my taste

Our response: In the previous round of revision, we reduced the length of the introduction by 20%. We worry that further reducing the introduction will compromise clarity. As the other reviewers did not raise additional concerns we feel it unwise to make further changes at this stage.

3) The GDF-15 hypothesis was a dead end and should go into the supplementary data.

Our response: We are reluctant to relegate the piece of data to supplementary information. It is commonly held that p53 mediates GDF15 signalling⁴⁻⁸, our data provide an important line of evidence for p53-independent activation of GDF15 in response to DNA damage. Given the substantial basic, pharma, and clinical interest in GDF-15 we feel it is important that this data, although negative is presented to the field.

Reviewer #3 (Remarks to the Author):

The authors have addressed the majority of my suggestions. I recommend that this article should now be published.

Our response: We are glad that our revisions have addressed the reviewer previous concerns and they now consider it ready for publication. We thank the reviewer for improving the manuscript during the process.

References

1. Dolle, M.E. et al. Broad segmental progeroid changes in short-lived Ercc1(-/Delta7) mice. *Pathobiol Aging Age Relat Dis* 1(2011).
2. Vougioukalaki, M. et al. Different responses to DNA damage determine ageing differences between organs. *Aging Cell* 21(2022).
3. Niederhofer, L.J. et al. A new progeroid syndrome reveals that genotoxic stress suppresses the somatotroph axis. *Nature* 444, 1038-43 (2006).
4. Cheng, J.C., Chang, H.M. & Leung, P.C. Wild-type p53 attenuates cancer cell motility by inducing growth differentiation factor-15 expression. *Endocrinology* 152, 2987-95 (2011).
5. Osada, M. et al. A p53-type response element in the GDF15 promoter confers high specificity for p53 activation. *Biochem Biophys Res Commun* 354, 913-8 (2007).
6. Tsui, K.H. et al. Growth differentiation factor-15: a p53- and demethylation-upregulating gene represses cell proliferation, invasion, and tumorigenesis in bladder carcinoma cells. *Sci Rep* 5, 12870 (2015).
7. Rochette, L., Meloux, A., Zeller, M., Cottin, Y. & Vergely, C. Functional roles of GDF15 in modulating microenvironment to promote carcinogenesis. *Biochim Biophys Acta Mol Basis Dis* 1866, 165798 (2020).
8. Jiang, W.W. et al. Emerging roles of growth differentiation factor-15 in brain disorders (Review). *Exp Ther Med* 22, 1270 (2021).